# Prefrontal cortex state representations shape human credit assignment

Amrita Lamba[1], Matthew R Nassar[2,3], Oriel FeldmanHall[1,3]*

[1]Department of Cognitive Linguistic & Psychological Sciences, Brown University, Providence, United States; [2]Department of Neuroscience, Brown University, Providence, United States; [3]Carney Institute of Brain Sciences, Brown University, Providence, United States

**Abstract** People learn adaptively from feedback, but the rate of such learning differs drastically across individuals and contexts. Here, we examine whether this variability reflects differences in *what* is learned. Leveraging a neurocomputational approach that merges fMRI and an iterative reward learning task, we link the specificity of credit assignment—how well people are able to appropriately attribute outcomes to their causes—to the precision of neural codes in the prefrontal cortex (PFC). Participants credit task-relevant cues more precisely in social compared to nonsocial contexts, a process that is mediated by high-fidelity (i.e., distinct and consistent) state representations in the PFC. Specifically, the medial PFC and orbitofrontal cortex work in concert to match the neural codes from feedback to those at choice, and the strength of these common neural codes predicts credit assignment precision. Together this work provides a window into how neural representations drive adaptive learning.

## Editor's evaluation

This study provides convincing evidence that the fidelity of neural representations of task states is associated with assigning credit to these states. The topic is timely and the results are important for understanding the neural mechanisms of reinforcement learning. The manuscript will be highly relevant for readers interested in cognitive and decision neuroscience, as well as reinforcement learning.

*For correspondence: oriel.feldmanhall@brown.edu

Competing interest: The authors declare that no competing interests exist.

## Introduction

Imagine that you are applying for a job and receive varying feedback on your 'pitch' from your interviewers. Which aspects of your pitch bear repeating for future interviews when similar behaviors across different settings produce varying outcomes? This example encapsulates the inherent difficulty of accurately linking outcomes to specific actions, particularly when the causal structure of the world is unknown and decision-irrelevant outcomes occur in close temporal proximity. In such scenarios, humans and animals are thought to group action-outcome contingencies together based on causal cues that reflect states in the environment, allowing outcomes to be selectively linked to specific states (*Collins and Frank, 2013*; *Gershman et al., 2015*). However, because learners have yet to discover the underlying generative structure of outcomes, and because learning and memory systems are fallible, outcomes can be misattributed and spread to irrelevant states—a challenge known as the structural credit assignment problem (*Hamid et al., 2021*; *Sutton, 1984*). In these cases, information gets smeared in memory (*Vaidya and Fellows, 2016*) which can result in overgeneralization.

While conventional reinforcement learning (RL) algorithms typically assume perfect credit attribution for each outcome observed, in reality, human learners run the gamut of how well they are able to actually assign credit. Take for example, learning about how honest each interviewer is in our

example. On one end of the spectrum, an individual may be able to link discrete outcomes to each interviewer, thus learning specific value associations at the person (stimulus) level. On the other side of the spectrum, a learner may average value across all interviewers, instead attributing outcomes to one global state. These scenarios represent discrete learning profiles that can substantially shape behavior beyond the influence of classic learning parameters, such as the learning rate or the magnitude of prediction errors (PE). Little is currently known about how states are represented in the human brain during learning (e.g. at the stimulus level or generalized across cues), or how credit is then assigned to these specific states. Thus, an open question is how does the human brain represent and successfully bind observed outcomes to the appropriate causal state to solve the credit assignment problem?

Non-human animal research hints that the prefrontal cortex (PFC) might play an integral role in credit assignment by binding state and action-value representations (*Asaad et al., 2017*), which could then be reinforced through midbrain dopaminergic signals to selectively gate reward attribution (*O'Reilly and Frank, 2006*). Tracking state-contingent outcome history would also be critical to properly assigning credit, which is believed to be governed by the lateral orbitofrontal cortex (lOFC; *Chau et al., 2015*; *Jocham et al., 2016*; *Walton et al., 2010*). Indeed, humans with lesions in the lOFC exhibit reduced state-contingent reward learning (*Noonan et al., 2017*) and display a greater tendency to misattribute rewards to irrelevant causal factors (*Vaidya and Fellows, 2016*). More recent work shows that the medial PFC and lOFC jointly track latent states (*Schuck et al., 2016*) by leveraging surprise signals (*Nassar et al., 2019*), allowing for credit assignment to be performed for both experienced (*Akaishi et al., 2016*) and unobserved outcomes (*Boorman et al., 2021*; *Witkowski et al., 2022*).

While prior work across species suggests that the PFC is involved in representing task states, it is not known whether the configuration of neural patterns play a role in credit assignment success. For example, unlike an eligibility trace, it is possible that learners actively represent task-relevant states during learning, enabling selective binding of outcomes to states in memory. Allowing temporally disparate actions and outcomes to be neurally bound to the relevant state representation likely supports increased discrimination between cue-specific decision policies. If this were the case, the precision of credit assignment may be contingent on the format and fidelity of state representations in the PFC (i.e. the degree of distinctiveness of each representation). Thus, to effectively guide credit assignment, a distinct neural code representing the state when feedback is delivered should then be evoked during a subsequent related choice. Failures to properly encode a state during choice or feedback should therefore result in increased misattribution and credit spreading (i.e. attributing outcomes more diffusely to irrelevant states).

In the current study, we leverage a computational neuroimaging framework by combining RL models with representational similarity analysis (RSA) to investigate whether distinct forms of credit assignment can be distinguished from neural patterns in the PFC. Our modeling framework allows us to estimate the precision of credit assignment from behavior, while RSA allows us to directly measure the content and fidelity of evoked neural state representations during choice and feedback. We further link the fidelity of these neural representations to how well an individual assigns credit across different learning contexts, allowing us to identify how credit assignment mechanisms are tailored to a particular situation. For example, during social exchanges humans are often able to exploit social feedback to quickly learn the value of social partners (*Lamba et al., 2020*; *van Baar et al., 2022*), which suggests that credit may be assigned selectively to specific individuals. In contrast, when learning in less familiar environments with an unknown causal structure (e.g. gambling with slot machines) learners may assign credit less precisely by spreading credit across states. Prior work also suggests that humans differ in how they strategically use feedback when learning about social partners compared to learning about reward-matched bandits (*Chang et al., 2010*; *Lamba et al., 2020*), providing an ideal empirical setup to probe for differences and commonalities in credit assignment precision across contexts.

Participants played an iterative, multiplayer social learning task which requires participants to distinguish between trustworthy and untrustworthy partners when making strategic monetary decisions, as well as a matched nonsocial gambling task with one arm bandits. We developed a credit assignment RL algorithm to capture the degree to which outcomes are precisely attributed to specific states at the stimulus level (i.e. to specific partners and bandits) or more generally to a single task state (i.e. across partners and bandits) through credit spreading. We find that different learning profiles are due to how precisely individuals assign credit, with some participants consistently spreading credit,

particularly when gambling and after a negative outcome. Multivariate neural patterns in prefrontal regions, including the lOFC and mPFC, encode state representations, but do so less precisely in those who spread credit. Indeed, high-fidelity state representations were associated with greater task earnings and more precise credit assignment—an effect that could not be explained by learning rate or the strength of the PE. Precise credit assignment is achieved through neural state representations sharing a common geometry across choice and feedback, signifying a persistent neural code indexing a stimulus' identity.

## Results

### Humans are faster to implement payoff maximizing strategies in the social domain

Participants (N=28) completed 60 trials of the Trust Game and a matched bandit task (order counterbalanced), while undergoing functional neuroimaging (fMRI). Participants made a series of monetary decisions with partners and bandits that varied in their reward rate (*Figure 1*, A to D) and could optimize their earnings by investing the full $10 with the high return stimulus and investing $0 with the low return stimulus. Despite being perfectly matched across social and nonsocial conditions, participants invested more money with the high return social partner compared to the bandit (mean social investment: $7.12; mean bandit investment: $6.43; $t=-3.57$, $p<0.001$; *Figure 2A*) and less money with the low return partner vs. bandit (mean social investment: $1.71; mean bandit investment: $2.71; $t=5.69$, $p<0.001$). No differences in mean investments were observed for the neutral or random stimuli across tasks (all ps >0.1).

### Investments are influenced by history of prior outcomes

To shed light on the credit assignment problem, we used a series of time-lagged regression models to examine how participants use relevant and irrelevant outcomes to guide learning. We paired each stimulus with its previous outcome and modeled the effect of three stimulus-matched (i.e. relevant) prior interactions on current investments (*Figure 1D*). The most recently experienced relevant outcome exerted the largest effect on investments (significant difference between slopes at t-1 vs. t-2: $t=-4.98$, $p<0.001$; *Figure 2B*). We also observed a stronger effect of previous outcomes on investments with partners compared to bandits (a significant effect at t-1 in which slopes were larger for the social vs. nonsocial task: $t=3.19$, $p=0.004$; *Figure 2B*). To investigate whether temporally close but irrelevant outcome history biased decisions, we yoked each investment to the immediately preceding outcome, irrespective of its identity. These outcomes should not inform choices on the current trial given the generative task structure. We observed that recent irrelevant outcomes (mean slope from t-1 through t-3) influenced decisions to gamble with bandits ($t=1.83$, $p=0.039$; *Figure 2B*), whereas recent irrelevant outcomes were anticorrelated with choices to trust partners ($t=-2.73$, $p=0.011$). These task differences in using irrelevant prior outcomes to guide choices were significant ($t=3.42$, $p<0.001$; *Figure 2B*). We also found a valence-dependent effect of outcome history on learning. Across tasks, prior relevant outcomes that were rewarding exerted a stronger influence on choices compared to losses (significant difference between slopes for gains versus losses; $t=-4.27$, $p<0.001$; *Figure 2C*). In contrast, immediately preceding irrelevant losses disproportionately impacted decisions compared to gains ($t=2.21$, $p=0.029$; *Figure 2C*), indicating that outcome misattribution is, in part, driven by losses. Put simply, learning from relevant outcomes was largely driven by gains, whereas outcome misattribution stemmed disproportionately from losses.

### Modeling credit assignment precision captures learning asymmetries

Given that we observed asymmetrical learning profiles across contexts, we probed whether divergent learning profiles were due to differences in credit assignment precision. We implemented a series of RL models using a continuous choice, logistic function algorithm with valence-dependent learning rates (V-LR; see Methods). In particular, we developed a credit assignment model which evaluates credit assignment precision along a continuum of perfect credit assignment (i.e. outcomes are correctly attributed to specific partner/bandit stimuli) to complete credit spreading across stimuli (i.e. outcomes are incorrectly attributed to a global state representing all partners/bandits). A credit assignment parameter (CA) evaluated the extent to which each PE selectively updates the expected

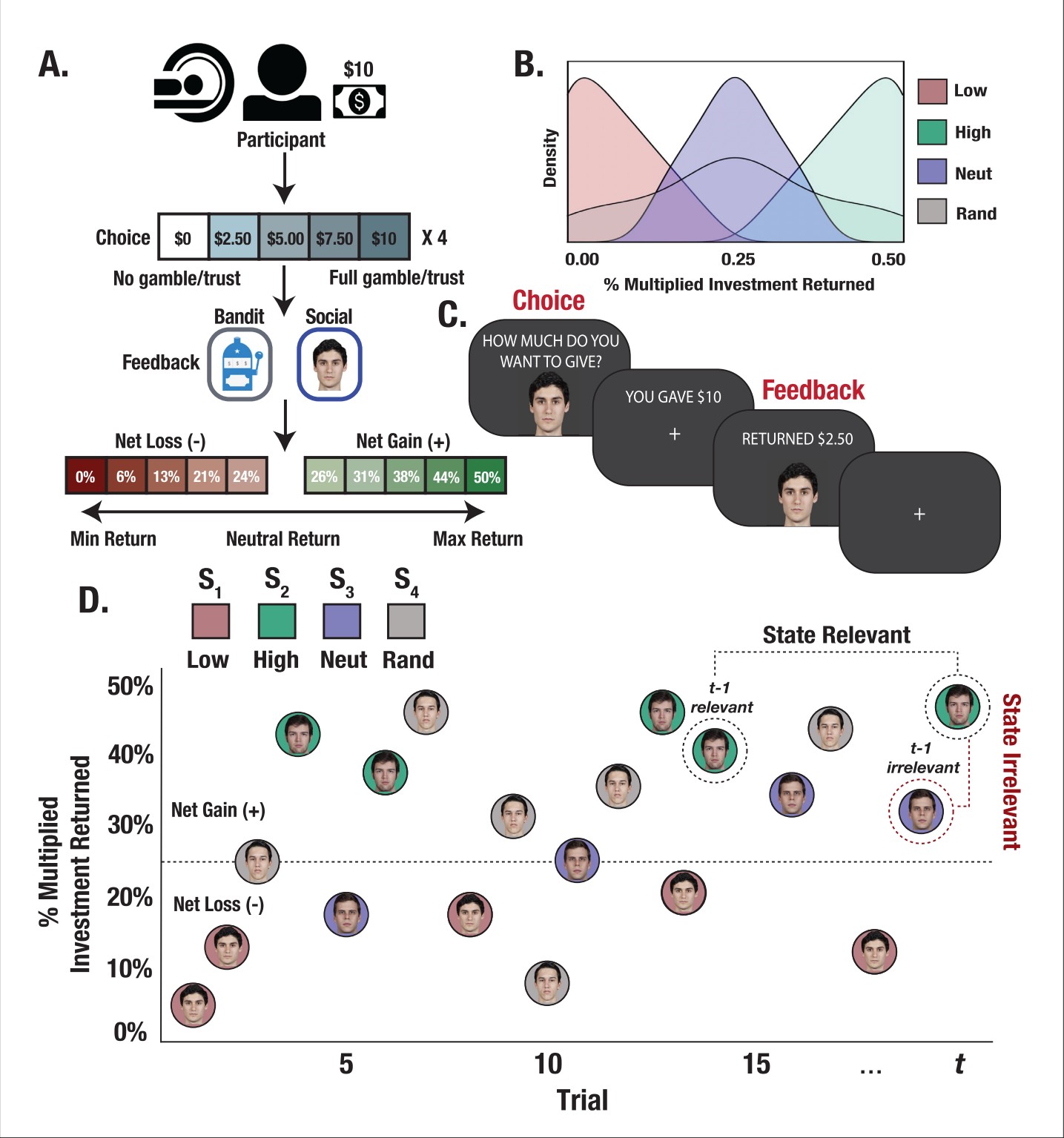

**Figure 1.** Experimental design and trial structure. (**A**) Trust Game and matched bandit task. Participants played 15 trials with each partner/bandit while in the scanner. On each trial, participants were paired with one of the four stimuli (partner/bandit) and given $10 to invest using a 5-button response box to indicate their investment in $2.50 increments. The monetary investment was then quadrupled, and partners/bandits returned anywhere from 0% to 50% of the money received, allowing for the possibility to double one's earnings, lose the full investment, or any outcome in between. (**B**) Task reward structure. Stimuli were randomly assigned to respond with fixed reward rates generated from one of four outcome distributions. Each stimulus deterministically returned less than the participant initially invested (low), more (high), an amount close to the initial investment (neutral), or a random

*Figure 1 continued on next page*

amount. (**C**) Task event sequence. Participants were given up to 3 s to indicate their choice, after which they experienced a jittered inter-stimulus delay. The returned investment was then displayed on the screen for a fixed 2 s duration. (**D**) Within-task stimulus presentation. Trials were randomly interleaved such that interactions with each stimulus could occur anywhere from 1 to 15 trials apart, allowing us to probe learning effects from relevant versus temporally adjacent irrelevant outcomes.

value of the relevant stimulus or updates the expected value of all stimuli concurrently (*Figure 3A*). Our model set included: (1) a baseline RL model that uses the Rescorla-Wagner update rule (see Methods), (2) a credit assignment model that includes a CA parameter controlling the degree to which outcomes affect the expected value of irrelevant states (V-LR, CA model), (3) a two parameter credit assignment model in which CA parameters were fit separately for positive versus negative PEs (V-LR, V-CA model), and (4) a model in which learned values decay gradually on each trial, such that errors were modeled through forgetting as opposed to a credit spreading mechanism (V-LR, Decay model). Behavior was best fit by the credit assignment model with valenced CA parameters (V-LR, V-CA) in both the trust and bandit task (see Methods). Consistent with the behavioral analyses above, our model reveals that people assigned credit more precisely to partners, and spread credit more diffusely across bandits (trust task mean CA estimate = 0.74; bandit task mean CA estimate = 0.54; *F*=10.67, p=0.002; *Figure 3B*)—an effect that was heightened for gains compared to losses (*F*=4.72, p=0.033; *Figure 3C*). Thus, our best fitting model revealed that different learning profiles across tasks were linked to credit assignment precision.

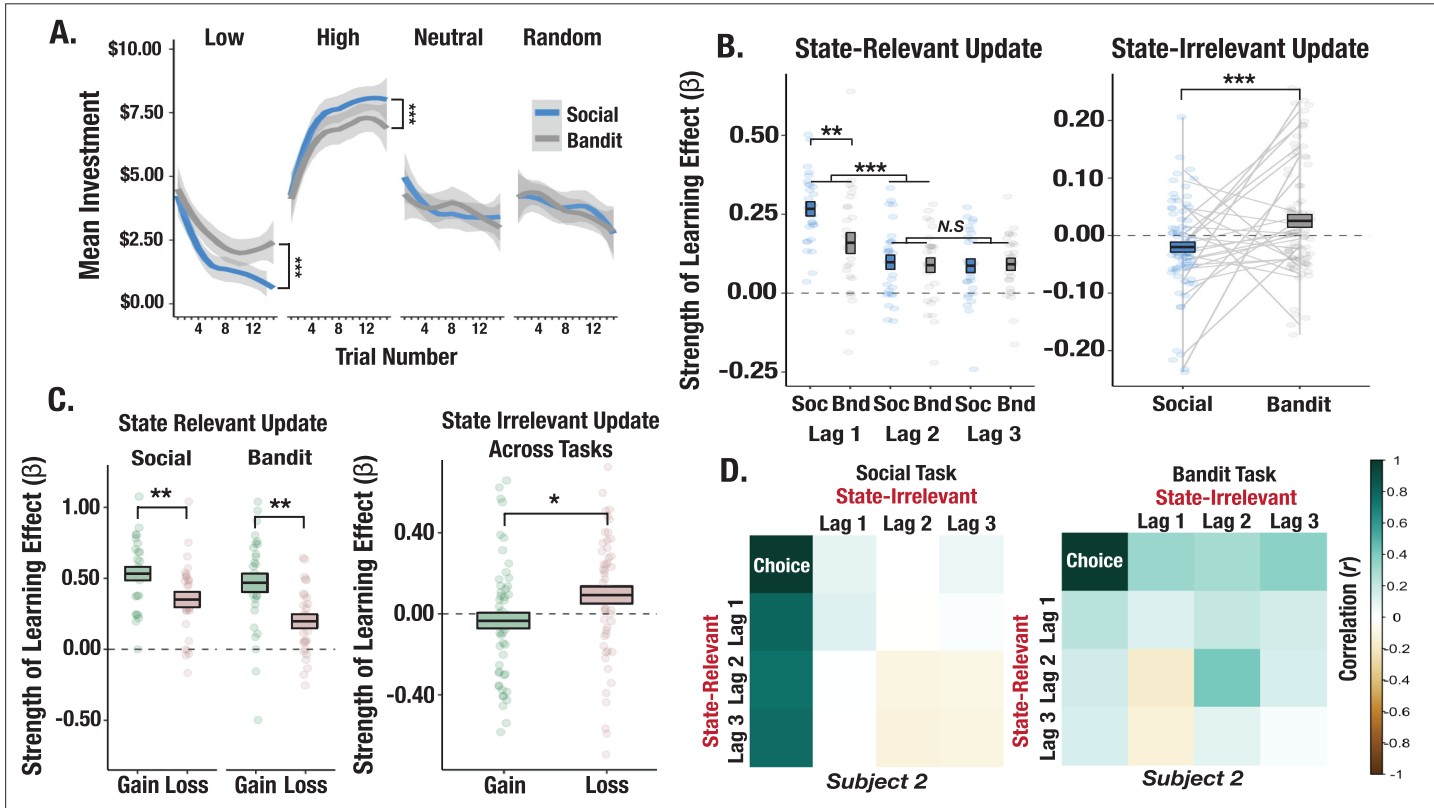

**Figure 2.** Behavioral differences across social and bandit tasks. (**A**) Learning curves from social and bandit tasks. Predicted investments over trials computed from fixed-effects regression model show faster learning in the social task. Shaded gray regions correspond to the standard error of the mean of the regression line. (**B**) Effect of relevant and irrelevant outcome history on choice. Model terms show increased learning from the most recent relevant outcome in the social task and an increased effect of irrelevant outcomes on investments in the bandit task. The box-length denotes the standard error of the mean, and the black line corresponds to the mean beta estimate for the lag term. (**C**) Effect of valence-dependent outcome history on choices. Relevant prior gains compared to losses exerted a greater influence on investments. (**D**) Correlation matrix of relevant and irrelevant outcomes on investments for a prototypical participant. The participant shows a strong pattern of learning exclusively from relevant outcomes in the social task but applies irrelevant outcomes to learning in the bandit task. Asterisks (*,**,***) denote p<0.05, p<0.01, p<0.001, respectively.

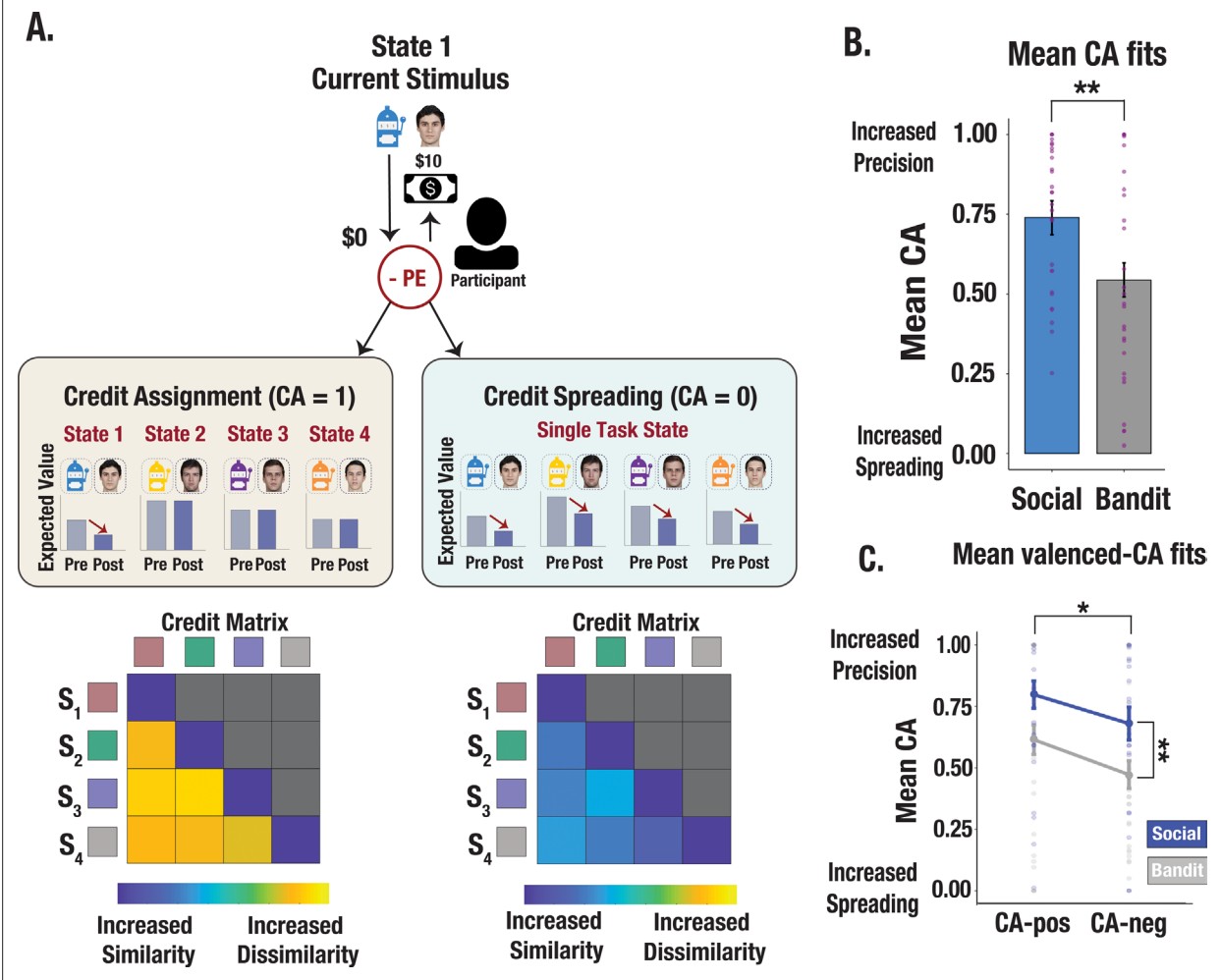

**Figure 3.** Illustration of credit assignment model and task differences in credit assignment precision. (**A**) Schematic visualization of credit assignment and credit spreading mechanisms. In a perfect credit assignment scenario (left side), PEs only update the expected value of the current state. A hypothetical credit matrix shows how credit assignment may impact the discriminability between states. Credit assignment values closer to 1 (perfect credit assignment) only use PEs from the relevant state to update expected values, therefore allowing for increased differentiation in the state space. Conversely, in a credit-spreading scenario illustrated with a hypothetical matrix (right side), PEs are used to update the expected value of current and irrelevant states, as if stimuli were clustered into a single causal state. This would result in less differentiation between states and increased confusability. (**B**) Credit assignment parameter estimates from the V-LR, V-CA model across social and bandit tasks. Mean CA fits show more precise credit assignment in the social task and increased spreading in the bandit task. Purple dots show individual parameter estimates and error bars denote the standard error of the mean. (**C**) Valenced credit assignment parameter estimates from the V-LR, V-CA model. Parameter fits from our valenced CA model show more precise credit assignment for gains (CA-pos) and more spreading for losses (CA-neg). Error bars show the standard error of the mean computed from a sample size of N = 28.

The online version of this article includes the following figure supplement(s) for figure 3:

**Figure supplement 1.** Continuous choice, logistic RL algorithm.

**Figure supplement 2.** Model performance and comparison.

**Figure supplement 3.** Model identifiability.

**Figure supplement 4.** MLE predictive check.

**Figure supplement 5.** Parameter recovery.

## Credit assignment predicts the fidelity of state representation during choice

Identifying and characterizing the neural circuitry involved in state representation can further clarify how credit assignment is implemented. We test the prediction that credit assignment requires a high

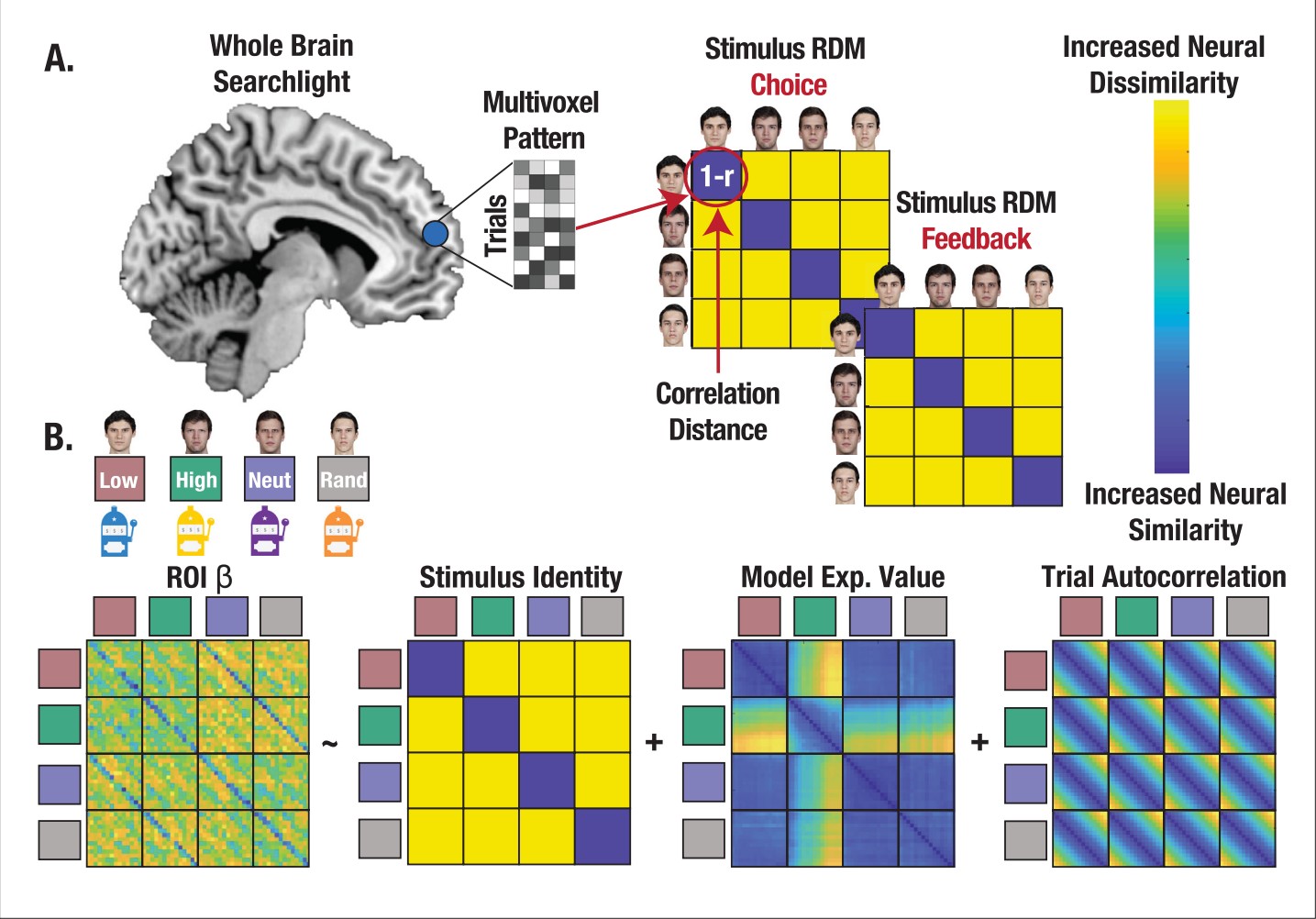

**Figure 4.** Representational similarity analysis (RSA) captures the format of state representations. (**A**) Conceptual depiction of RSA methods with whole brain searchlight. Multivoxel patterns were extracted for all trials and reorganized into a correlation distance (1 r) matrix with trials nested within stimulus identities for each task. State representation was evaluated separately for choice and feedback. (**B**) Regression approach estimating state representation in neural ROIs, controlling for expected value (**V**) and trial autocorrelation.

fidelity (i.e. distinct and consistent) neural representation of a stimulus' identity during choice, which should in theory support retrieval of the state-specific decision policy (NB: while we use *state representation* to describe the process, precise credit assignment is observed when a specific *stimulus' identity* is associated with a discrete outcome). Using a whole brain searchlight, we extracted single trial coefficients of the neural pattern on each trial and for each stimulus to create a neural representational dissimilarity matrix (RDM), separately for choice and feedback (*Figure 4A*; see Methods). We then computed the correlation distance between each searchlight RDM and our identity hypothesis matrix, controlling for additional regressors in regions of interest (ROIs; separate ROIs constructed from choice and feedback searchlights) that survived correction for multiple comparisons (*Figure 4B*; see Methods). This enabled us to evaluate the degree to which neural patterns differed across stimulus' identities, while also measuring the extent to which neural patterns were consistent across trials, thus serving as a high-fidelity 'stamp' of the stimulus identity for credit assignment.

Neural patterns in a constellation of brain regions including the mPFC, lOFC, and mOFC met basic criteria for providing state representations (statistically significant beta coefficients of the identity RDM) at the time of choice (*Figure 5B*; see *Supplementary file 1a* for ROI coordinates). Within these regions, stimulus' identity was more strongly encoded in the trust vs. bandit task across ROIs (trust task mean $\beta$=0.019; bandit task mean $\beta$=0.012; $t$=−3.64, p<0.001; *Figure 5B*), consistent with the findings from our model. There was also a positive relationship between an individual's mean CA

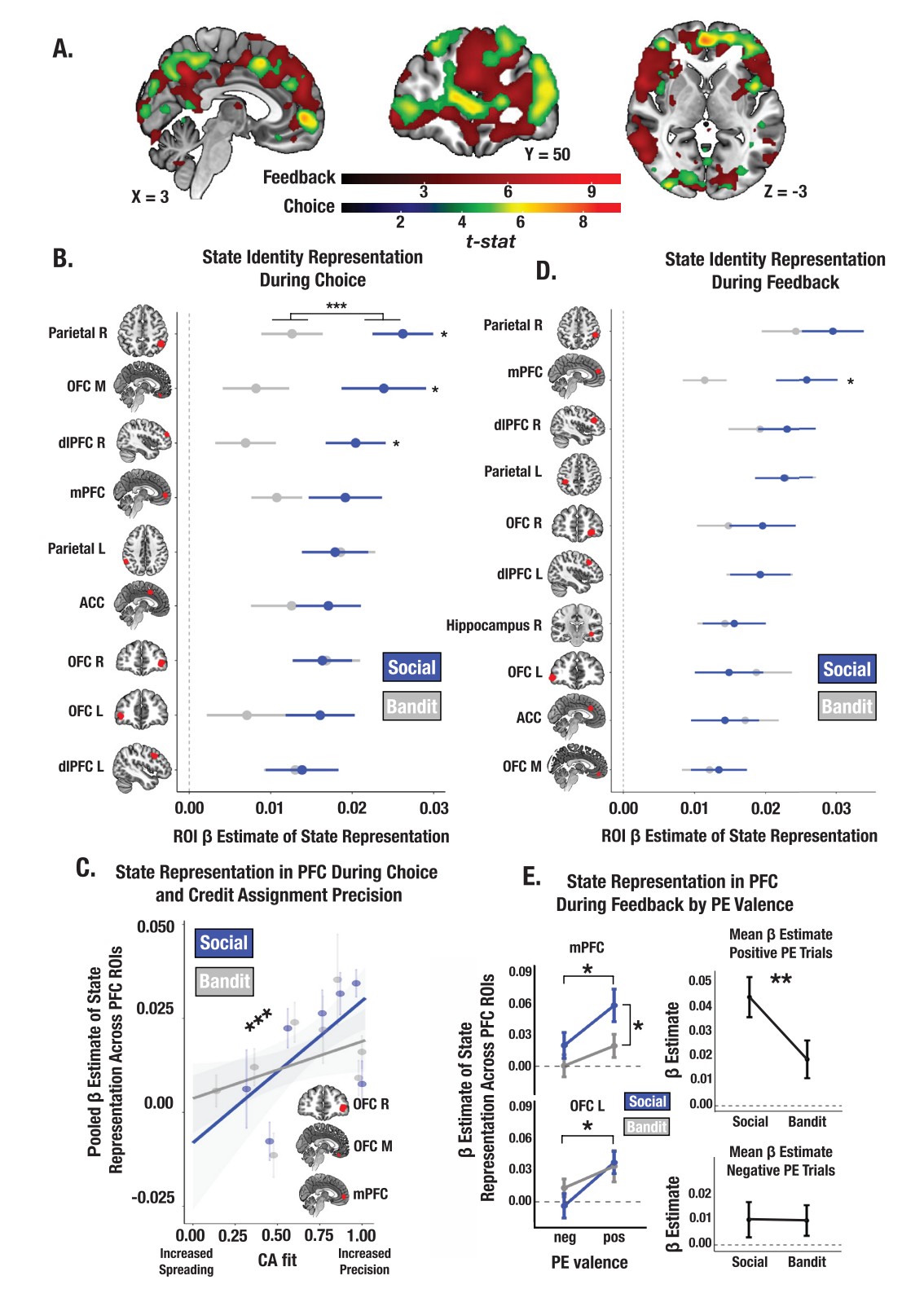

**Figure 5.** Social and bandit task state representations during choice and feedback. (**A**) Task-summed group-level t-map of state representation during choice and feedback; image is thresholded at the cluster-level ($P_{FWE}$ <.05) and at peak level (*P*<.0001). (**B**) Beta estimates of state representation across ROIs identified from choice phase searchlight, broken down by social and bandit tasks. RSA results indicate stronger state representation in the social task. Individual asterisks denote a significant within-subject effect for the specified ROI. (**C**) Predictive association between individual CA parameter

*Figure 5 continued on next page*

*Figure 5 continued*

estimates and the fidelity of state representation in mPFC, lOFC, and mOFC during choice. Across both social and bandit tasks, credit assignment predicts the strength of state representation in the PFC. *** Denotes the main effect of CA on the pooled estimate of state representation across ROIs (p<0.001). (**D**) State representation estimates in ROIs during feedback. Task differences only emerge in the mPFC. (**E**) Effect of PE valence on state encoding in the PFC during feedback. Across social and bandit tasks, positively valenced PEs were associated with higher-fidelity state representations in the mPFC and lOFC. Pooling across mPFC and lOFC ROIs, state encoding was greater in the social compared to bandit task on positive PE trials. Error bars show the standard error of the mean computed from a sample size of N = 28.

parameter estimates and the strength of the stimulus' identity representation in the PFC ROIs across both tasks ($t$=3.38, p<0.001; *Figure 5C*) which did not depend on outcome valence (CA positive: $t$=3.26, p=0.0014; CA negative: $t$=2.50, p=0.014)—revealing that individuals who assigned credit more selectively to the relevant stimulus' identity also had more consistent and distinct neural representations of it. Furthermore, the more money earned in the task, the more discriminable these stimulus representations were in the PFC ROIs in both tasks ($t$=4.14, p<.001). These results accord with the prediction that the fidelity of state representations in the PFC during choice control the precision of credit assignment by supporting increased differentiation between state-specific decision policies.

## Reward enhances encoding of state representations during feedback

At the time of feedback, a stimulus' identity must be sufficiently encoded so that outcomes can be linked to the appropriate prior action. Using our searchlight approach, we identified a suite of regions providing state representations during feedback (*Figure 5D*; see *Supplementary file 1b* for ROI coordinates). We observed stronger identity representations in the trust task in the mPFC ($t$=–2.41, p=0.023; the fidelity of state representations did not differ across tasks in any additional ROIs). We then examined whether positive vs. negative PEs differentially modulate the strength of state encoding during feedback. We estimated the strength of stimulus' identity encoding within our prefrontal ROIs separately for positive and negative PE trials for each participant (Methods). Across tasks we observed stronger identity encoding during positive vs. negative PE trials in the mPFC (main effect of valence: $F$=4.33, p=0.039; *Figure 5E*) and lOFC ($F$=5.98, p=0.017), an effect that was significantly greater in the trust compared to bandit task ($t$=–2.71, p=0.008). Together, this suggests that valence asymmetries in credit assignment precision (i.e. more precise credit assignment for rewards, and increased credit spreading for losses) emerge because reward enhances the strength of state representations in the PFC, especially in social contexts.

## Successful credit assignment hinges on shared representational geometry between choice and feedback

How is information from choice and feedback integrated to support learning? We consider the possibility that credit assignment is achieved by neurally binding a specific outcome to certain stimuli. One way in which this may happen is by matching the identity representations from the last relevant outcome to the next relevant choice. Given the observed involvement of the mPFC and lOFC, these brain regions are likely candidates for being able to match neural codes across time. The idea is that greater alignment of neural representations across choice and feedback can preserve a common neural code of the stimulus' identity, supporting increased credit assignment precision. To examine the shared representational structure between these timepoints and the extent to which increased alignment reflects a common identity representation, we identified conjunction ROIs from voxels that survived permutation testing in both the choice and feedback searchlight analyses (*Figure 6A, B*). We then computed the degree of neural pattern similarity between the representations at choice and feedback in these PFC ROIs, and evaluated the degree to which the shared geometry preserves information about the stimulus' identity (Methods). Across both tasks, we observed a significant positive relationship between an individual's credit assignment precision (CA parameters) and the consistency of their representations of the stimulus across both timepoints in the mPFC and lOFC (pooled estimate across ROIs: $t$=2.17, p=0.033; *Figure 6C*; see *Supplementary file 1c* for ROI coordinates)—an effect that was selectively enhanced for gains but not losses ($t$=3.48, p<0.001). Thus, the precision of credit assignment, particularly for rewarding outcomes, was associated with increased neural binding between feedback and choice—a process supported by shared geometry of stimulus' representation across distinct phases of learning.

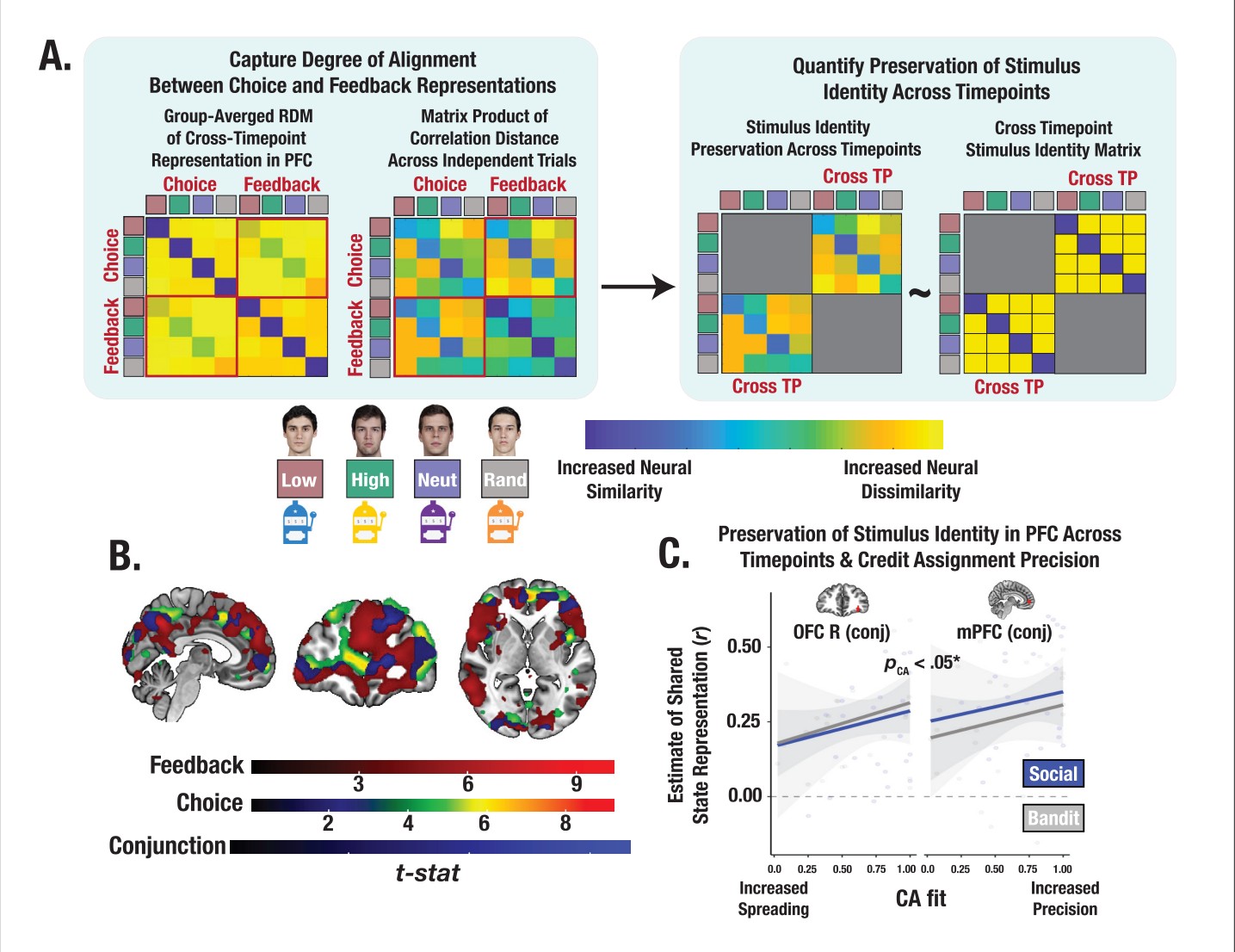

**Figure 6.** Cross-timepoint representational similarity analysis (RSA). (**A**) Conceptual depiction of cross-timepoint RSA. For each participant, cross-timepoint matrices were constructed as the correlation distance between even and odd trial neural RDMs (Methods). Cross-timepoint cells from the matrix were selected and then correlated with a cross-timepoint identity matrix to estimate the degree to which the shared structure within the neural patterns across choice and feedback reflected a stimulus' identity. (**B**) Task-summed group-level t-maps displaying results of conjunction contrasts (Methods). Group-level image is thresholded at the cluster-level ($P_{FWE}$ <.05) and at peak level (p<0.0001). (**C**) Predictive association between individual credit assignment estimates and the consistency of identity representational structure in mPFC and lOFC (i.e., conjunction ROIs) across choice and feedback. * Denotes the effect of CA on the pooled estimate of shared state representation across ROIs (p<0.05).

## The strength of learning signals cannot explain credit assignment precision

Given that we observed individual differences in credit assignment precision across tasks, we tested whether these differences could be explained by the strength of PE signaling. Our results could be simply explained by the magnitude of PE signaling in the social task. To rule this out, we conducted a whole brain parametric modulation to test the relationship between trial level PEs from our V-LR, V-CA model and changes in the amplitude of the BOLD signal (see Methods). A task-summed *t*-map revealed significant clusters in the ventral striatum and ventral medial prefrontal cortex (vmPFC; corrected for multiple comparisons; *Figure 7A*; see *Supplementary file 1d* for ROI coordinates). Critically, the strength of PE signaling did not differ across the trust and bandit tasks in these regions (all ps >0.1; *Figure 7B*), even at a lowered threshold (p<0.001 uncorrected). We also wanted to rule

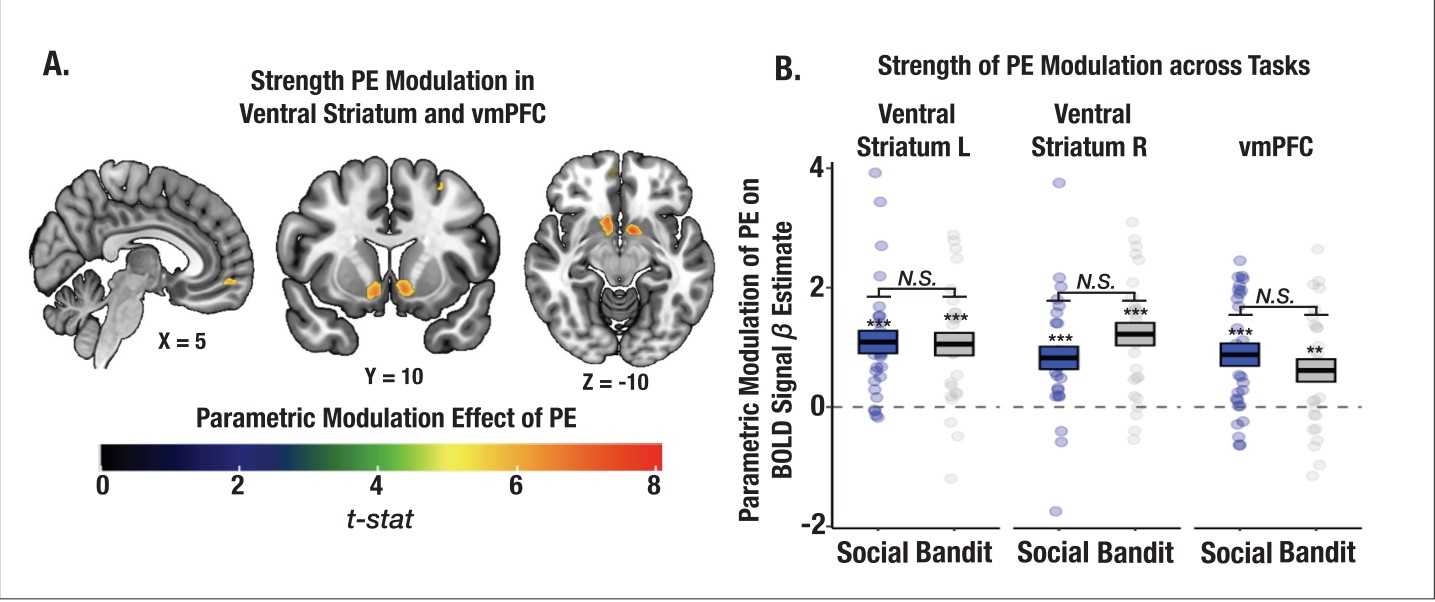

**Figure 7.** Strength of PE signaling across social and bandit tasks using parametric modulation analysis. (**A**) Task-summed group-level t-map displaying the parametric modulation effect of PEs on the BOLD signal (Methods). Significant clusters were observed in the ventral striatum and vmPFC. Group-level image is thresholded at the cluster-level ($P_{FWE}$ <.05). (**B**) Despite observed learning differences in the two tasks, the magnitude of PE modulation did not differ between social and bandit tasks.

out the possibility that CA precision might simply be attributed to learning rate (LR) differences. We interrogated whether the influence of PEs on the expected value of each state predicted the format of stimulus representations in the PFC, either at the time of choice or in the cross-timepoint representation. The effect of CA on stimulus representation was significant even when controlling for the learning rate (choice: $t$=3.31, p<0.0012, cross-timepoint: $t$=2.11, p=0.036; both LR and CA were included as additive terms predicting choice and cross-timepoint beta coefficients for state representation), and LR was not significant in either model (all ps >0.09). Together, these findings support the claim that it is not the overall magnitude of learning signals that shape the specificity of learning, but rather, how these learning signals are attributed to specific states through credit assignment.

## Discussion

Adaptable learning systems—whether human, animal, or artificial—must be able to exploit causal structure by differentiating between spatial and temporal cues, allowing learners to balance behavioral flexibility with specificity (*Asaad et al., 2017*; *Soto et al., 2014*; *Tenenbaum et al., 2011*; *Walton et al., 2010*). Structural credit assignment in particular, enables learners to integrate contingencies into a set of learned decision policies from predictive features and cues in the environment (*O'Reilly and Frank, 2006*; *Sutton, 1984*). Here we show *how* humans achieve successful structural credit assignment. First, people are more precise when attributing outcomes to other people than they are to slot machines, an effect that is enhanced for gains compared to losses. Second, the PFC assigns credit by matching neural codes across time. State representations must be initially sufficiently encoded during feedback, a process that is more robustly observed in the gain, compared to loss, domain. High fidelity state representations during subsequent choice also support increased state discrimination and improved learning specificity. The critical lynchpin for how precisely credit is assigned in the PFC is the degree to which an individual exhibits a shared neural geometry in state representations between feedback and choice. Put simply, the PFC serves as a hub for credit assignment by leveraging reward signals and organizing state representations into a shared neural code, allowing for the efficient transfer of credit from feedback to subsequent choice. This functional coordination between distinct phases of learning aligns with prior work demonstrating that lOFC and mPFC track task states (*Schuck et al., 2016*), particularly those that govern the structure of rewards, so that humans can

respond flexibly and adaptively to shifting environmental demands (*Jocham et al., 2016*; *Nassar et al., 2019*; *Witkowski et al., 2022*).

We further observed that an outcome's valence asymmetrically shapes the specificity of assigning credit. More precise outcome attribution was observed for gains while increased credit spreading was observed for losses. Although past studies offer insight into credit assignment for rewards (*Asaad et al., 2017*; *Hamid et al., 2021*; *Jocham et al., 2016*; *Walton et al., 2010*), no studies that we are aware of have documented asymmetrical effects of PE valence driving the precision of credit assignment. Thus, these results offer an interesting parallel to existing stimulus generalization theories. Prior work using classical conditioning paradigms, in which a neutral stimulus is paired with an aversive outcome, shows that the conditioned response is also evoked by novel stimuli (*Dunsmoor and Paz, 2015*; *Hull, 1943*; *Schechtman et al., 2010*). Transfer effects to a new stimulus follows a generalization gradient in which the experienced intensity of the prior aversive outcome predicts increased threat generalization (*Dunsmoor et al., 2017*; *Lissek et al., 2005*), and greater perceptual generalization (*Davis et al., 2010*; *FeldmanHall et al., 2018*). Although our paradigm articulates generalization through credit spreading mechanisms, from a signal detection standpoint, these findings undoubtedly dovetail with that notion that it is 'better to be safe than sorry' in the aversive domain (*Dunsmoor and Paz, 2015*). This may help to explain why increased credit spreading of negative PEs across irrelevant cues can become pathological and maladaptive, offering potential inroads into understanding the etiology of generalized anxiety disorders.

While this work unveils a generalizable computational and neural mechanism for structural credit assignment, there are a few limitations and a number of unanswered questions that future work can help address. First, our study did not control for the visual complexity of the stimuli across tasks (i.e. human faces compared to colored bandits). While it is possible that the observed differences in learning and state decoding reflects the salience of faces in the trust task, our prior work suggests that this account is unlikely, given that when the stimuli's complexity was perfectly matched across social and nonsocial tasks, we still observe faster learning in the social domain (*Lamba et al., 2020*). An open question revolves around the valence-asymmetric credit assignment effects, which may have interesting mappings onto dopamine modulation in the striatum and amygdala. Recent work in mice finds that wave-like dopamine signals from the dorsal striatum communicate when successful actions performed during instrumental learning are necessary for performance, offering insight into the underlying neuromodulatory dynamics of credit assignment in the reward domain (*Hamid et al., 2021*). Conversely, prior work suggests that dopamine modulation in the amygdala gates the selectivity of an acquired threat response, whereas inhibition of amygdala dopamine receptors is linked to threat overgeneralization (*De Bundel et al., 2016*). These findings are consistent with our results that the strength of PE signaling alone does not sufficiently explain differences in learning specificity. Future work should consider how midbrain dopamine modulation in the striatum and amygdala mechanistically interact to shape the format of downstream state representations.

Notably, our whole brain searchlight also picked up state representations in other prominent cortical networks, such as the control network (lateral parietal, anterior cingulate, and dorsal lateral prefrontal regions), and may accord with the possibility that distinct functional networks encode abstract state representations that vary only in format to optimize for differing task demands (*Vaidya and Badre, 2022*). Future work could consider how prefrontal and control networks interact during successful credit assignment, particularly when abstract state representations are required to perform complex sequences of actions. To summarize, our results identify a simple and domain-general neural mechanism for credit assignment in which outcomes and states are temporally bound together in the PFC, revealing a biologically grounded model for how humans assign credit to causal cues encountered in the world. How this mechanism coordinates with other known neural systems and deviates in psychiatric disorders has yet to be uncovered.

## Methods

### Participants

Data was collected from 30 right-handed adults (ages 21–36; mean age = 23.5, $N_{female}$ = 16) in the Providence, Rhode Island area. Our study protocol was approved by Brown University's Institutional Review Board (Protocol #1607001555) and all participants indicated informed consent before

completing the social and the bandit tasks in the scanner. After all fMRI preprocessing steps were completed, two participants were removed from the final sample due a high degree of motion artifacts (movement >3 mm). All participants received monetary compensation ($15 /hour) and additional performance-based bonus payment of up to $20.

## Instructions and stimulus presentation

Prior to scanning, all participants were given instructions for the social and bandit tasks and instruction ordering was counterbalanced depending on which task participants completed first in the scanner. For the social task, participants were told they would see the faces of previous participants who had already indicated the proportion of the investment they wished to return and who had been photographed prior to leaving their session. In reality, each of the four face stimuli were drawn from the MR2 database (*Strohminger et al., 2016*). All face stimuli included in our task were prejudged to be equivalent on trustworthiness and attractiveness dimensions by independent raters (*Strohminger et al., 2016*). Slot machine stimuli varied by visually distinct colors (purple, blue, yellow, and orange).

For each task, participants were required to pass a basic comprehension check to ensure that they understood the payoff structure of each game. Stimuli were presented using Psychtoolbox in MATLAB 2017a. Each trial was designed to elapse over a 16 s duration. The trial was initiated with a choice phase with a 3 s response window in which participants indicated their investment using a 5-button response box (options: $0, $2.50, $5.00, $7.50, $10.00). After participants keyed in their response, a jittered interstimulus interval (ITI), randomly distributed between 1 and 5 s, reminded participants of their investment. The outcome was then presented for a fixed 2 s duration, following by an additional ITI, filled with however many seconds remained for the full 16 s trial duration (between 6 and 13 s). If participants failed to indicate their investment within the 3 s response window, the investment was considered $0, and a missed trial prompt appeared during the ITI. Missed trials were omitted from all behavioral and RSA analyses. Stimulus ordering was randomly interleaved, and therefore consecutive presentations of the same stimulus could occur anywhere from 1 to 15 trials apart following a right-skewed distribution, such that most consecutive stimulus interactions occurred within 1–5 trials.

## Reward structure

Each stimulus in the social and bandit tasks was randomly assigned to follow one of four reward distributions, such that stimulus identities were counterbalanced across different payoff structures. The high reward stimulus always returned more than the participant initially invested and thus always resulted in a net gain, whereas the low reward stimulus always returned a lower amount resulting in a net loss. Neutral and random stimuli were designed to return a roughly equivalent amount and served as a control for outcome valence, allowing us to examine outcome attribution precision with stimuli that resulted in net gains and losses with equal frequency. Neutral and random stimuli only differed in terms of their return variance (i.e. the extremity of gains and losses). Rather than truly sampling from a payoff distribution which could have resulted in vastly different observed outcomes across participants and tasks (e.g. observing a consistent string of gains or losses on the extreme end of the distribution simply due to chance), we preselected the return rates so that the full range of the distribution was sampled from. We then applied these preselected return rates for all participants and in each task but allowed their ordering to be randomized across trials. Notably, although return rates were fixed, the payoff on each trial was still dependent on the participant's investment (see task structure in *Figure 1*).

## Time-lagged behavioral regression analyses

Single trial investments were modeled using a regression that included the monetary amount returned on previous trials at various time points (i.e. lags) as explanatory variables. To capture individual learning effects, we modeled investment data for each participant separately including both social and bandit tasks in the same linear regression model. Investments were modeled as a weighted sum of previous returns experienced on relevant trials (i.e. those with a matched stimulus) as well as irrelevant trials (i.e. those that immediately preceded a given investment, irrespective of stimulus identity). We fit slopes for the contribution of each lag term using previous returns from the $n$th trial back in each task (social vs. bandit), yielding 13 coefficients per participant (model equation below; $i,t$ denotes each participant and trial, respectively).

$$\text{Investment}_{(i,t)} = \beta_0 + \beta_{1,2}\text{Lag}_1\text{Rel}_{(i,t)}|\text{task} + \beta_{3,4}\text{Lag}_2\text{Rel}_{(i,t)}|\text{task} + \beta_{5,6}\text{Lag}_3\text{Rel}_{(i,t)}|\text{task} +$$

$$\beta_{7,8}\text{Lag}_1\text{Irrel}_{(i,t)}|\text{task} + \beta_{9,10}\text{Lag}_2\text{Irrel}_{(i,t)}|\text{task} + \beta_{11,12}\text{Lag}_3\text{Irrel}_{(i,t)}|\text{task}$$

To model whether rewards vs. losses differentially impacted reward attribution, for each participant we separated trials into scenarios in which the previous stimulus-matched and previous irrelevant outcome resulted in a net gain (return >investment) or a net loss (return <investment). Here we considered only lag1 trials to minimize parameter tradeoffs that prevented model convergence, and furthermore ran four separate regression models quantifying the effects of lag1 returns for each combination of relevant vs. irrelevant and gain vs. loss trials.

## Logistic reinforcement learning model

To better understand trial-to-trial changes in investing, we developed a nested set of RL models to translate trial-outcomes into behavioral updates. Because choices in the task were both discrete and ordinal in their magnitude (choice options: $0, $2.50, $5.00, $7.50, $10.00), we used a logistic function to model the learned value of investing with each partner/bandit type based on trial and error.

## Model investment function

In each of our models, all choices in which the participant responded were included in model fitting. Predicted investments for each trial were generated from a sigmoid function that included parameters to account for individual investment biases (i.e. baseline differences in investment preferences) and the slope of the relationship between $V_t$ and predicted investments (m; *Figure 3—figure supplement 1A*), where $V(t,j)$ reflects the expected value for investing in stimulus $j$ (i.e. a specific partner/bandit) on trial $t$:

$$\text{pred. investment}_t = \frac{\text{max investment}}{1 + e^{-m(V_{(t,j)} - \text{bias})}}$$

To get the likelihood with which our model would produce all possible investments on a given trial, we assumed that the probability of a given investment would fall off according to a Gaussian probability density function (PDF) around the predicted investment (*Figure 3—figure supplement 1B*):

$$p\left(\text{investments}\right) = \frac{1}{\sigma\sqrt{2\pi}}e^{\frac{-\left(\text{all investments} - \text{pred.investment}\right)^2}{2\sigma^2}}$$

The width of the Gaussian distribution was fixed to a value of 1 in all models, controlling the variability in model investments. The probability of investing—*p*(investment)—generated from the Gaussian PDF was normalized on each trial such that the total probability across investments was equal to 1, and the model was fit by minimizing the negative log of the sum of *p*(investments) corresponding to the actual participant investments across trials.

## Modeling learning

We fit a learning model to behavior separately for social and bandit tasks. In our baseline model, we used the Rescorla-Wagner learning rule (*Rescorla, 1972*) to compute the reward prediction error ($\delta$) on each trial ($t$), which updated the expected value ($V$) of investing with each partner/bandit type ($j$) after each outcome observation. Error-driven learning was then scaled by the learning rate ($a$):

$$\delta = \text{reward}_t - V_{(t-1,j)}$$

$$V_{(t,j)} = V_{(t-1,j)} + a \cdot \delta$$

We used a matrix to store the updated value of $V$ on each trial separately for each stimulus, resulting in a trial × stimulus $V$ matrix for each participant and each task. Most of our models included a prior parameter (see *Supplementary file 1e*), which estimated the initial value of $V$ in the first row of the $V$ matrix. If a prior parameter was not included in the model, the $V$ matrix was initialized at 0. In the baseline model, outcomes were always attributed to the appropriate state (i.e. updated $V$ for only the appropriate stimulus), effectively performing standard model-free RL. We included additional learning rate, credit assignment, and decay parameters to the base model to construct a set of models

that varied in complexity, and that provided distinct conceptual accounts of learning differences, but notably all models mapped experienced outcomes to learned values to guide future choice (see *Supplementary file 1e* for a full list of models tested and *Supplementary file 1f* for a description of each parameter).

To capture individual differences in credit assignment vs. credit spreading, we introduced a credit assignment parameter (CA) that quantifies the degree to which observed outcomes on the current trial influenced the expected value of irrelevant causes (i.e. all other partners/bandits not engaged with on the current trial), denoted with the index $k$.

$$V_{(t,j)} = V_{(t-1,j)} + a \cdot \delta \cdot CA$$

$$V_{(t,k)} = V_{(t-1,k)} + \frac{a \cdot \delta \cdot (1 - CA)}{n_k}$$

To account for valence-dependent learning effects, we fit models with valenced learning rate (V-LR) parameters, in which PEs greater or less than 0 were scaled by positive or negative learning rates, respectively (V-LR, CA model). We also fit a model with valenced CA terms (V-CA) to capture valence-specific differences in reward attribution (i.e. whether credit assignment vs. spreading is dependent on observing better or worse than expected outcomes). This model (V-LR, V-CA) was algorithmically identical to the V-LR, CA model, except that two separate CA parameters were used to account for credit assignment on positive and negative PE trials.

## Decay models

We modeled forgetting effects based on a decay model previously described in *Collins and Frank, 2012* in which a decay parameter gradually adjusts learned values back to initial ones (i.e. the prior), proportionally to the degree of forgetting. In our model (V-LR, Decay), decay ($\gamma$) and the prior were fit as additional free parameters to each participant.

$$V_{(t,j)} = V_{(t-1,j)} + a \cdot \delta$$

$$V_{(t,k)} = V_{(t-1,k)} + \gamma \cdot \left( prior - V_{(t-1,k)} \right)$$

## Model comparison

Model fits were then evaluated using the Akaike information criterion (AIC), which we computed as:

$$AIC = -2 \left( BayesLL \right) + 2 \left( n \ parameters \right)$$

We performed model selection by maximizing the negative AIC and minimizing Δ AIC, which was calculated as the difference between each participant's best-fitting model and every other model in the set. This approach allowed us to evaluate model performance penalized for additional terms and the model fit advantage of each participant's best-fitting model relative to every other model in the set. Thus, the best-fitting model would ideally be able to explain each participant's data approximately as well, if not better, in most instances. Model comparison was performed separately for social and bandit tasks (see *Supplementary file 1g-1h* and *Figure 3—figure supplement 2* for AIC and Δ AIC values of each model).

## Model validation

Model confusability was evaluated by simulating 100 participants per model. For each simulated participant, free parameters were randomly sampled from a uniform distribution and trial-to-trial investments were generated under the sampled parameterization. Each model was fit to each simulated participant using 20 iterations of gradient descent, and classification rates were computed as the frequency with which each participant was best fit by the correct generative model, which we evaluated by maximizing the negative AIC (rate of winning model/true model; see *Figure 3—figure supplement 3A*). The inverse confusion matrix (*Figure 3—figure supplement 3B*) was based on the same data but shows the probability that each generative model gave rise to a given 'best fitting' model. We also performed a maximum likelihood estimation check (i.e. posterior predictive check) for the set of MLE-optimized parameters from the V-LR, V-CA model. Model generated data

shown in *Figure 3—figure supplement 4* captures empirical patterns. Parameter recovery, shown in *Figure 3—figure supplement 5*, indicates that parameters from the V-LR, V-CA model were reliably estimated and recoverable.

## MRI data acquisition

Data was acquired at the Brown University MRI Facility with the Siemens Prisma 3T MRI Scanner. Anatomical scans were collected using a T1-weighted sequence with 1 mm³ isotropic voxels, 1900ms TR, flip angle = 9 degrees, 160 slices/volume, 1 mm slice thickness, for a duration of 4 min, 1 s. Functional scans were acquired using a T2-weighted sequence with 3 mm³ isotropic voxels, 2000ms TR, flip angle = 78 degrees, 38 slices/volume, 3 mm slice thickness. Each task was divided into 2 functional runs, each consisting of 246 volumes, for a duration of 8 min and 12 s. We used a bounding box with a forward tilt along the AC-PC axis to ensure we were imaging lateral and medial OFC.

## Data preprocessing

Data was preprocessed in SPM12. For multivariate analyses, data was preprocessed in the following sequence: slice-time correction, realignment, co-registration, segmentation, normalization, spatial smoothing. Images were normalized to a standard MNI template and resampled to 2 mm³ voxels. For univariate parametric modulation analyses, we used the same preprocessing sequence, except images were realigned prior to slice-time correction.

Images were smoothed using a 2 mm³ smoothing kernel for multivariate analyses and with an 8 mm³ kernel for univariate analyses. RSA images constructed from the deconvolved time-series GLM (see below) were later smoothed using a 6 mm³ smoothing kernel for group analyses.

## Time-series GLM

For each participant we obtained time-series estimates of the BOLD signal by deconvolving the HRF using single trial regressors (*Mumford and Poldrack, 2007*; *Ramsey et al., 2010*), concatenated across task runs. We also included separate trial regressors for choice and feedback onsets within each GLM (i.e., choice and feedback onsets were modeled simultaneously) to control for potential temporal correlations in the BOLD signal resulting from consecutive task events. For choice phase regressors, we modeled voxel activations during the choice duration, which occurred within a 3 s window. Feedback phase activations were modeled during a fixed 2 s duration. We included six motions regressors derived from realignment, along roll, pitch, yaw and x,y,z dimensions to control for motion artifact. We also included additional framewise displacement (FD) regressors, using a FD threshold = 1.2 to identify noisy images. We then regressed out noise from frames above the FD threshold, including the previous images and subsequent two images, based on recommendations (*Power et al., 2012*; *Power et al., 2014*).

## Whole brain searchlight analysis

We conducted four independent searchlight analyses across the whole brain to measure representational dissimilarity during choice and feedback phases of the task, and separately for social and bandit tasks. For each participant and each task phase (choice and feedback), we first selected the relevant trial regressors (e.g. choice regressors for all trials in which the participant responded). We then constructed a 9 mm radius spherical searchlight, which we moved along x,y,z coordinates of participant-specific brain masks (binary mask of voxels with sufficient accompanying BOLD activations), with a step size of 1, such that the center of the searchlight was placed in each voxel once.

We then extracted single event (choice/feedback) beta coefficients from all voxels within the searchlight from the relevant phase regressors modeled in our deconvolved time-series GLM and noise normalized the coefficients (*Walther et al., 2016*). Beta coefficients from each phase regressor were then extracted and reorganized into a voxel by trial coefficient matrix (*Figure 4A*). To align the searchlight neural RDM with our state identity hypothesis matrix, we reorganized the coefficient matrix by nesting trial within each stimulus type. To obtain the searchlight RDM, we then computed the correlation distance (1 *r*) between each row and column element in the coefficient matrix. Correlation distance values from the lower triangle (all elements off the identity line) of our neural RDM were then z-transformed and correlated with the lower triangle of our z-transformed predictors, using the following linear model to obtain a *t*-statistic estimating the effect of state identity for each searchlight

(neural correlation distance ~state identity +autocorrelation term). We examined the effect of the state identity and expected value hypothesis matrices separately in our whole brain searchlight analyses to avoid any systematic correlations that could potentially emerge from predictor collinearity but allowed state identity and expected value predictors to compete in our ROI analyses. The resulting first-level $t$-maps from the whole brain searchlights for choice and feedback and for each task were then spatially smoothed with a 6 mm$^3$ Gaussian smoothing kernel before being submitted to second-level analyses.

To avoid constructing task-biased ROIs, we created summed $t$-maps from each participant's social and bandit searchlights using SPM's imcalc function (images created using a simple summation method; social $t$-map +bandit $t$-map). We then conducted second-level analyses on the task-combined $t$-maps for choice and feedback phases of the task. To correct for multiple comparisons, we conducted non-parametric permutation testing on the second-level analyses, using a cluster-forming threshold of p<0.0001 and a null distribution based on 5000 permutations. Permutation testing was conducted with the SnPM package (*Hayasaka and Nichols, 2003*). We then created binary masks of all voxels in our second-level analyses that were significant at the cluster-level ($p_{FWE}$ <.05) and at the peak level (p<0.0001) and used the task-combined corrected $t$-maps to identify ROIs. We used a data driven approach to identify two sets of ROIs for choice and feedback by extracting coefficients from statistically significant cluster peaks in corrected $t$-maps. We limited the cluster size of our ROIs to be no larger than the size our searchlight by placing a 9 mm radius sphere at the center of the local maxima and extracted all voxels from the sphere that survived permutation testing. Within each ROI from our two sets chosen from choice and feedback searchlights, coefficients from the model were then disaggregated to independently evaluate the strength of state representation in the social vs. bandit task (shown in *Figure 5*, B to C), which we computed from the following model (neural correlation distance ~state-identity RDM + expected value RDM + autocorrelation term).

## Valence-based RSA

For each participant, trial-level deltas (PEs) estimated from the valenced-ca model using the MLE optimized parameters were z-scored to control for the extremity of experienced gains and losses across participants, to ensure we had sufficient trials from each stimulus within both positive and negative valence RDMs, and to ensure a roughly equivalent amount of data in each neural RDM. The z-transformed PEs were then used to separate data into the appropriate RDM and our RSA procedure for each ROI was then separately applied for positive and negative RDMs and for choice and feedback phases.

## Cross-timepoint RSA

Prior to computing the cross-timepoint correlations across task phases, we constructed a set of ROIs in candidate areas that were involved in state representation during both choice and feedback. To avoid constructing ROIs that were biased towards a particular task phase, we integrated task-combined $t$-maps (choice $t$-map +feedback $t$-map). We then masked our task and phase-combined image using a binary conjunction $t$-map of voxels that survived permutation testing in *both* our choice and feedback phase analyses, using a 9 mm radius sphere centered at the local peaks to isolate voxels from statistically significant clusters and focusing specifically on voxels in the PFC. We then computed the cross-timepoint correlation within each of our conjunction ROIs, separately for each task.

Our approach for computing the cross-timepoint correlation was to first create two data sets for each participant by separating the data into even and odd trials that occurred with each partner/bandit type (e.g. set 1: all even trials with low, high, neutral, random stimuli, set 2: all odd trials with low, high, neutral, random stimuli), so that representational correspondence could be measured across consecutive stimulus-matched trials that were temporally distanced in time (trial ordering was interleaved such that stimulus-matched trials were always 1–15 trials apart in the task; *Figure 1D*). We then constructed a data matrix for each of our even and odd data sets, including choice and feedback in same matrix. For each matrix, we extracted beta coefficients from each voxel on each trial within even and odd data sets, and concatenated choice and feedback patterns. For both even and odd data sets, we then averaged beta coefficients in each voxel, resulting in two voxel × stimulus matrices reflecting the average beta in each voxel at choice and feedback timepoints (rows) and activations for each stimulus during choice and feedback (columns). We then calculated the correlation distance (1 $r$)

between even and odd matrices. In the cross-timepoint quadrants of the resulting matrix, mean activations on even trials during the choice (choiceEven) were compared to mean activations on odd trials during the feedback (feedbackOdd), and even trials during feedback (feedbackEven) were compared to odd activations during choice (choiceOdd). The resulting matrix product of even and odd RDMs thus allowed us to evaluate the shared structure of state representations across task phases (the lower left and upper right quadrants of the cross-timepoint RDM; *Figure 6A*) and across independent trials, therefore breaking any systematic temporal or autocorrelation signals within the neural pattern. We then correlated only the cross-timepoint quadrants of the matrix product with a cross-timepoint state identity matrix to quantify the degree of representational alignment in the neural code across choice and feedback timepoints that preserved the state identity. The degree of shared information in the neural code (quantified as Pearson correlation coefficients) was subsequently submitted to further regression analyses to evaluate the association between shared representational geometry and credit assignment precision (*Figure 6C*). We constructed ROIs in the mPFC and lOFC, given that clusters in these regions emerged from our conjunction *t*-map and signaled cross-timepoint state encoding.

## Parametric modulation analysis

We conducted univariate parametric modulation analysis to evaluate the effect of trial PEs on the amplitude of the BOLD signal, allowing us to capture differences in the strength of PE coding across tasks. For each participant, we constructed a first-level design matrix that included regressors for both choice and feedback onsets and durations, and trial-level PEs from our V-LR, V-CA model. Social and bandit task data were modeled in the same GLM, and we included additional regressors for motion (see time-series GLM section). In group analyses we then generated a *t*-contrast against 0 for the PE regressor. This contrast included data from both social and bandit tasks so that we were blinded to task condition when selecting ROIs. To correct for multiple comparisons, we performed permutation testing on the task-combined *t*-map (see procedure in the whole brain searchlight analysis section). We then selected ROIs from significant cluster peaks that survived permutation testing and extracted beta coefficients from ROIs separately for social and bandit tasks.

## Acknowledgements

We thank Michael J Frank for comments and feedback and Eric Ingram for assisting with data collection. We also thank Avinash Vaidya for helpful RSA code and discussion. This work was supported by NARSAD grant 26210 to OFH.

## Additional information

### Funding

| Funder | Grant reference number | Author |
|---|---|---|
| Brain and Behavior Research Foundation | 26210 | Oriel FeldmanHall |

The funders had no role in study design, data collection and interpretation, or the decision to submit the work for publication.

### Author contributions

Amrita Lamba, Conceptualization, Data curation, Software, Formal analysis, Investigation, Visualization, Methodology, Writing – original draft, Project administration, Writing – review and editing; Matthew R Nassar, Software, Formal analysis, Supervision, Methodology, Writing – review and editing; Oriel FeldmanHall, Conceptualization, Formal analysis, Supervision, Funding acquisition, Methodology, Writing – review and editing

### Author ORCIDs

Amrita Lamba ⓘD https://orcid.org/0000-0002-8703-2886
Matthew R Nassar ⓘD https://orcid.org/0000-0002-5397-535X
Oriel FeldmanHall ⓘD https://orcid.org/0000-0002-0726-3861

### Ethics

Our study protocol was approved by Brown University's Institutional Review Board (Protocol #1607001555) and all participants indicated informed consent for both behavioral and neuroimaging portions of the study.

### Decision letter and Author response

Decision letter https://doi.org/10.7554/eLife.84888.sa1
Author response https://doi.org/10.7554/eLife.84888.sa2

---

## Additional files

### Supplementary files

Supplementary file 1. Additional tables with fMRI analysis ROI coordinates, cluster size, and peak statistics, and additional information about model parameters and performance. (a) Choice Phase ROI coordinates. (b) Feedback Phase ROI coordinates. (c) Conjunction ROI coordinates from cross-timepoint analysis. (d) Parametric modulation ROI coordinates. (e) List of RL Models included in model comparison, including their respective free parameters indicated with the ×. V denotes valenced terms for either the learning rate (LR) or credit assignment (CA) parameters in the model. (f) Logistic RL algorithm parameters indicating model behavior at upper and lower bounds. (g) Mean AIC and SE for each model. Mean AIC for V-LR, V-CA was the max in the set. Mean values are plotted below in *Figure 3—figure supplement 2*. (h) Model Comparison. Model Comparison was performed by minimizing Δ AIC, which was computed as the difference between each participants best-fitting model and each model in the set (see Methods). Δ AIC was the lowest for the V-LR, V-CA model indicating that the model captured the behavioral data better than other models in the set, and in instances in which a participant's data was better fit by another model the V-LR, V-CA model could explain the data equally as well. Δ AIC with individual points is plotted below in *Figure 3—figure supplement 2*.

MDAR checklist

### Data availability

Behavioral data, analyzed neural data, and code are available on github: https://github.com/amrita-lamba/eLife_prefrontal_credit_assignment. (copy archived at *Lamba, 2023*).

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
