## [Editor Report]

This study provides convincing evidence that the fidelity of neural representations of task states is associated with assigning credit to these states. The topic is timely and the results are important for understanding the neural mechanisms of reinforcement learning. The manuscript will be highly relevant for readers interested in cognitive and decision neuroscience, as well as reinforcement learning.

---

## [Decision Letter]

**Decision letter after peer review:**

Thank you for submitting your article "Prefrontal cortex state representations shape human credit assignment" for consideration by *eLife*. Your article has been reviewed by 2 peer reviewers, one of whom is a member of our Board of Reviewing Editors, and the evaluation has been overseen by Christian Büchel as the Senior Editor. The following individual involved in the review of your submission has agreed to reveal their identity: Rui Ponte Costa (Reviewer #2).

Apologies for the long delay in getting back to you. The reviewers have now discussed their reviews with one another, and the Reviewing Editor has drafted this to help you prepare a revised submission.

Essential revisions:

As you can see, both reviewers agreed that your manuscript addressed an important and timely topic. However, they also identified key weaknesses that should be addressed with substantial text revisions, new analysis, and/or data.

The list below highlights two key concerns, but please also consider the points raised in the individual critiques.

1. Potential stimulus confounds (reviewer 1 public review, point 1). Ideally, you would address this with new data showing that the results hold when the stimuli between the social and non-social tasks are matched. Alternatively, you could change the language about social vs. non-social to complex vs. simple in most of the manuscript, and refer to the potential role of social vs non-social stimuli in the discussion.

2. Are effects driven by the strength of neural representations rather than learning signals (reviewer 2 public review, point 3)? Please address this concern through additional analysis as it is central to your main conclusions.

*Reviewer #1 (Recommendations for the authors):*

This study uses fMRI and computational modeling to examine the relationship between credit assignment and neural stimulus representations, and compare this relationship between yoked social and non-social reinforcement learning tasks. The authors find that credit assignment is more accurate in the social task and that this is mirrored in the strength of neural stimulus representations in the PFC. They also report evidence for overlapping neural representations between the choice and feedback phases.

The question addressed in this study is timely and interesting, and the manuscript is well-written. However, there are several shortcomings in the experimental design and analytic approach that limit what can be concluded from these results.

1. A focus of the study are differences between social and non-social tasks. The key distinction between these tasks is the stimulus set: the social task uses faces as stimuli whereas the non-social task uses a bandit symbol in different colors. This potentially confounds the social vs non-social learning domain with the salience, complexity, etc. of the stimuli used. Thus, it is unclear whether the behavioral and neural results reflect credit assignment in social vs. non-social domains or the well-known effects of stimulus features on learning.

2. The authors use computational modeling to compare different mechanisms of learning (working memory vs credit assignment). However, although these two mechanisms are not mutually exclusive and there is no attempt to capture both mechanisms within the same model.

3. The authors use the term 'state' when referring to different stimuli within each task. This is misleading because 'states' are typically conceptualized to be abstract and not tied to specific stimuli. The analyses presented in the manuscript do not dissociate state identity from stimulus identity, and so it would be more accurate to refer to 'stimulus identity' rather than 'state identity' in the context of the current manuscript.

4. My biggest concern is related to potential stimulus confounds in the social vs non-social tasks. Faces are more complex, salient, and meaningful than the colored bandits. We know from decades of learning and memory research that such stimulus features determine the rate of learning and so any differences between conditions may have nothing to do with social vs non-social but are entirely driven by these features.

Relatedly, given the choice-related RSA results reflect representations of stimulus identity, any differences between the two conditions could be driven entirely by better decoding of more complex stimuli.

There are two solutions to this issue. Either you do additional experiments and show that a reasonable set of features does not explain differences between the two tasks, or you revise the manuscript to remove all language about social vs non-social and acknowledge that the differences between the tasks could be entirely driven by differences in stimulus features.

5. Your modeling approach is designed to test (among other things) whether working memory or credit assignment mechanisms better account for the behavioral data. Given that these mechanisms are not mutually exclusive, it would be important to include a model that captures both mechanisms.

6. I think using the term 'state' when referring to 'stimulus' is misleading. States are generally thought to be abstract and not tied to specific stimulus configurations. To dissociate states and stimuli, you'd need an experiment where the same state is evoked by different stimuli. Your design actually allows you to do that if you assume that faces and corresponding bandits evoke the same state.

7. I may be wrong here, but if I understand your description of the cross-timepoint RSA in the method section correctly, I wonder whether this analysis could still be confounded by the temporal proximity of choice and feedback phase. As far as I understand, you computed the similarity of choice and feedback phases within odd trials and separately within even trials. Then you computed the matrix product between the odd and even trial neural RDMs. Couldn't any signals (including those driven by vascular effects or noise) that linger from the choice phase into the feedback phase explain the cross-phase similarity within the odd and even trials? It seems it would be better to compute two neural RDMs such that the choice phase comes from odd trials and the feedback phase comes from even trials (and vice versa), and then average the two RDMs. This way, the similarity between the choice and feedback phase is less likely to result from autocorrelations

8. How were the ROIs selected? I assume there were more areas that represented stimulus identity, no?

*Reviewer #2 (Recommendations for the authors):*

In this manuscript the authors use a combination of reinforcement learning modelling and fMRI to study the fidelity of state representations in the frontal cortex and whether these representations predict the ability of (individual) participants for credit assignment.

Strengths:

1. This study provides a nice combination of reinforcement learning modelling and fMRI studies, which enabled the authors to link computational principles with neuronal representations.

3. The experimental paradigm is also interesting, contrasting social and non-social tasks. It suggests very interesting differences between the two in terms of investment, but also in terms of neural representations.

4. Finally, this study might make substantial advances in our understanding of individual differences in terms of credit assignment, a critical part of learning and adaptability. However, this is not entirely clear (see below).

Weaknesses:

1. The manuscript could present some of the results in a more gradual way, that makes it accessible to a general reader. I also find that there are generally long relatively complex sentences that make it hard to follow. For example: "We leverage these potential differences to evaluate whether the brain adaptively adjusts the fidelity of state representations across contexts to mediate the selectivity of causal learning.".

2. There is in my view a need to clarify what is meant by credit assignment (CA). For example, the way the models are described in the main text make it seem that some models perform credit assignment whereas others do not. From what I can see all these models have to perform some form of credit assignment, as they all atribute credit to model parameters/states, which I think would also make model 1 a CA model.

3. One of the key points made by the authors is that individual differences are driven by the strength of neural representations, and not by the magnitude of learning signals. Unfortunately, I fail to see how this conclusion is supported by the analysis of the data. I believe that this is interpretation builds on their correlation analysis between the representations and the model fits (Figure 5D and 6C). However, the model itself contains an explicit learning signal (δ: prediction error), so it is not clear to me how can the authors disentangle the neuronal representations from the learning signal using this analysis.

4. The model and some of the methods are not described in enough detail. For example, it is not stated what some of the parameters are. Although the models used appear to be standard in the field, no links/citations to classical RL models are made.

Claims:

Several claims are supported by the data/analysis, but it is not clear to me if one of the central claims is (see one of the weakness points above).

1. One of the potentially very interesting points made by the authors is about what causes individual differences in terms of CA. However, I fail to see how they can use the CA fit without implicitly also considering learning signals -- learning signals are an implicit part of their model (i.e. there is a δ = prediction error). Not sure how this can be addressed, but maybe there is something I'm missing and this is not a problem at all? On a related point, I failed to see a discussion on this very important part of the manuscript, discussing what would be the key contribution of the paper.

2. The text could be improved by making more smoother/gradual descriptions of the results/ideas. For example, I find that there is a lot of information in the final part of the section "Although these outcomes should not inform choices on the current trial given the generative task structure…" (until the end). To make it more accessible to the general reader I suggest that you present this information more gradually and provide intermediate (brief) summaries of what it means.

3. So I think it would be better to clarify that these are all different variants of CA models. This does not change your interpretation in any way but would make the story more clear. Also, to follow the modelling more easily I would suggest giving specific names to each model like you already do for the v-CA model. I note that this is done in Figure S2, but should also be included in the main text, and in the methods when referring to the methods, for clarity. This would also make the link with Figure 3 easier to follow. For example, at the moment it's unclear exactly to which model Figure 3b and c refer.

4. On page 7 and Figure 3 you use the term "spread" again, but this was only referred to very early on, so at this point appears out of place. Would be important to highlight what this means and how the models capture this spread. Also, in Figure 3 the spread is contrasted with CA precision, given that these two elements are critical for understanding Figure 3, it would be important to clearly define them in the main text before introducing the results of Figure 3. Also, in Figure 3 are the credit matrices obtained by running the models? If so, how? Clarify.

5. One possible prediction for the neuronal representations of PE found in PFC is that they should weaken over learning if they underlie the learning signal. As this form a critical component of any credit assignment/learning model, it would be interesting/important to explore this. I imagine this could be easily tested with your data by analysing how the PE-specific representations change over learning.

6. There is some lack of clarity in the description of the model/analysis: (i) The baseline RL model, seems to be exactly the Rescorla-Wagner rule. If so please refer to it as so. (ii) The AIC is usually given by: AIC = – 2ln(L) + 2k, not AIC = – 2ln(L) – 2k, as given in the methods. Please clarify. (iii) What is the prior in the decay model? It does not appear to be given. (iv) The Q variable is usually reserved for Q-learning in RL, which models explicitly the value of state-action pairs, but this does not appear to be the case here. Instead, to be consistent with the literature, I suggest that you use the variable V. It's important that the authors ensure that all the details are given, and parameters described.

---

## [Author Response]

Essential revisions:As you can see, both reviewers agreed that your manuscript addressed an important and timely topic. However, they also identified key weaknesses that should be addressed with substantial text revisions, new analysis, and/or data.The list below highlights two key concerns, but please also consider the points raised in the individual critiques.1. Potential stimulus confounds (reviewer 1 public review, point 1). Ideally, you would address this with new data showing that the results hold when the stimuli between the social and non-social tasks are matched. Alternatively, you could change the language about social vs. non-social to complex vs. simple in most of the manuscript, and refer to the potential role of social vs non-social stimuli in the discussion.2. Are effects driven by the strength of neural representations rather than learning signals (reviewer 2 public review, point 3)? Please address this concern through additional analysis as it is central to your main conclusions.Reviewer #1 (Recommendations for the authors):This study uses fMRI and computational modeling to examine the relationship between credit assignment and neural stimulus representations, and compare this relationship between yoked social and non-social reinforcement learning tasks. The authors find that credit assignment is more accurate in the social task and that this is mirrored in the strength of neural stimulus representations in the PFC. They also report evidence for overlapping neural representations between the choice and feedback phases.The question addressed in this study is timely and interesting, and the manuscript is well-written. However, there are several shortcomings in the experimental design and analytic approach that limit what can be concluded from these results.1. A focus of the study are differences between social and non-social tasks. The key distinction between these tasks is the stimulus set: the social task uses faces as stimuli whereas the non-social task uses a bandit symbol in different colors. This potentially confounds the social vs non-social learning domain with the salience, complexity, etc. of the stimuli used. Thus, it is unclear whether the behavioral and neural results reflect credit assignment in social vs. non-social domains or the well-known effects of stimulus features on learning.

Thank you for raising these critiques. First, while we are generally in agreement that the complexity of a stimuli’s’ features can affect the rate of learning, it is unlikely that complexity differences in our stimulus set can account for the credit assignment differences we observed. This is for the following reason: In our previous work (see Lamba, Frank & FeldmanHall, 2020, *Psychological Science*), we compared social and nonsocial learning using the same task setup (trust game vs. bandit task, albeit with gradual changes in reward dynamics over time), where the complexity of stimulus identities were perfectly matched: colored silhouettes represented social partners and colored slot machines represented bandits. Despite the social and nonsocial stimuli being matched in their visual complexity, participants learned to adapt their behavior to social partners more quickly than to bandits, which could be attributed to how feedback was being used to guide learning. In other words, in this prior data set, with a larger sample (N = 354) where the tasks contained visually matched social and nonsocial stimuli, we still observed learning differences between social and nonsocial contexts. We therefore decided to depart from the previous experimental design which controlled for the stimulus’ visual complexity in order to create a more realistic and engaging experiment in the current work—which is why social partners are represented by faces.

Second, the Reviewer states that *“a focus of the study are the differences between social and non-social tasks”.* This statement, combined with their concern about differences between stimulus complexity suggests that our motivation for using social and nonsocial testbeds was not adequately conveyed in the initial manuscript, which led readers to focus on differences between the social and nonsocial contexts as a main thrust of the paper. The goal of our study is to characterize how credit assignment unfolds in the human brain, not to make normative claims about which domains people are “better” at assigning credit. The use of social/nonsocial conditions was a manipulation to show that different learning contexts in general can impact how learning unfolds, which we link to the format of neural representations in the PFC. We have revised the introduction of our manuscript to make this framing clearer (page 3, lines 85-94) and have removed the word “accurate” when describing social versus nonsocial credit assignment differences.

In short, while we agree that stimulus complexity can be an important contributor to behavior, we believe this is unlikely to be driving differences in credit assignment between our two tasks due to evidence from our previous work that controlled for stimulus complexity and yet still documented learning differences. That being said, we now acknowledge on page 16, lines 376-381 of the manuscript that the stimuli are not matched on complexity, which may contribute to learning differences.

2. The authors use computational modeling to compare different mechanisms of learning (working memory vs credit assignment). However, although these two mechanisms are not mutually exclusive and there is no attempt to capture both mechanisms within the same model.

We agree that they are not mutually exclusive! We would ideally be able to use a single model that simultaneously captures both working memory and credit assignment. In fact, in an initial model we included credit assignment and decay parameters (to account for forgetting) within the same model. However, this model was not identifiable and produced high confusability with other models (i.e., the valenced Learning Rate (V-LR), Decay model and the V-LR, V-CA model) during identifiability tests (see Figure S1). Parameter fits from this model were also unstable, suggesting that simultaneously estimating decay and credit assignment parameters within the same model can be difficult. Given our task design, which only contains 15 trials for each stimulus and 60 trials per task, we wanted to avoid over-fitting and interpreting parameters from overly complicated models, which is why we did not include this model in our final set.

We also want to clarify that testing competing hypotheses between working memory and credit assignment interference was not a central aim of this work, and have therefore revised language in the modeling section of the manuscript (page 7, lines 185-187) to avoid generating this impression. Rather, we fit the decay model to rule out the possibility that forgetting was the *primary* mechanism driving differences in task performance, even if some lapses in working memory were of course going to be concurrent with credit spreading mechanisms. In our V-LR, Decay model, decay parameters did not significantly differ between task conditions *(t* = 1.86, *df* = 27, *p* = 0.074) and decay parameters did not predict the fidelity of state representations in the PFC at the time of choice (*t* = -0.462, *p* = 0.65) or the shared geometry between choice and feedback (*t* = -1.612, *p* = 0.11). In addition, decay parameter estimates from the V-LR, Decay model heavily clustered around 0, suggesting that for the most part, lapses in working memory were not fundamentally contributing to behavior—which alleviates some of our worry about omitting the decay parameter in our winning credit assignment model.

3. The authors use the term 'state' when referring to different stimuli within each task. This is misleading because 'states' are typically conceptualized to be abstract and not tied to specific stimuli. The analyses presented in the manuscript do not dissociate state identity from stimulus identity, and so it would be more accurate to refer to 'stimulus identity' rather than 'state identity' in the context of the current manuscript.

Thank you for raising this point—we think it is an important one to address on both a conceptual and scientific level. When writing the manuscript, we selected the word state to intentionally discriminate it from the term stimulus. We use the term state as defined in the reinforcement learning literature, that is, to mean the complete set of environmental variables relevant for defining a behavioral policy. Under this definition, the state of a tic-tac-toe game would include whether each box is filled with an X, O, or is empty (Sutton & Barto 2018). In our credit assignment model and task setup, we consider precise credit attribution to indicate linking outcomes to specific stimuli (i.e., partners/bandits), so in these cases states should be represented at the stimulus level (1 state=1 stimuli)—or as the reviewer points out, “stimulus identity”. In complex real-world problems, we contend that knowing the correct state space for reinforcement learning is not so simple, and that in some cases it might be perfectly reasonable to learn a policy that applies to all humans (or bandits). Extending this logic to our task would yield a situation where all stimuli map to a single “state”. As we show in our paper, participants that spread credit do not learn at the stimulus level and instead seem to learn about a single task state that encompasses *all* stimuli. Indeed, in our model, credit assignment values of 0 perform RL with exactly one state. Since our credit assignment parameter was continuous, we can capture this tendency along a continuum of representing states at the level of a stimulus, or, at the level of a single state (in which all stimuli are grouped together), and we also see this reflected in the format of neural representations in the PFC. We apologize that our justification of this terminology was not adequately conveyed in the initial manuscript and have revised the paper to make this logic clearer to readers (page 2, lines 39-45; page 7, lines 175-180; Figure 3A, and Page 9 lines 214-216). In line with the reviewer’s recommendation, we have also revised the manuscript throughout to (1) make it clear what our analyses are interrogating (fidelity of the stimulus identity); and (2) when an individual is learning at the state level (failing to learn about individual stimuli) versus learning at the stimulus level, in which case we use the term “stimulus identity”.

4. My biggest concern is related to potential stimulus confounds in the social vs non-social tasks. Faces are more complex, salient, and meaningful than the colored bandits. We know from decades of learning and memory research that such stimulus features determine the rate of learning and so any differences between conditions may have nothing to do with social vs non-social but are entirely driven by these features.Relatedly, given the choice-related RSA results reflect representations of stimulus identity, any differences between the two conditions could be driven entirely by better decoding of more complex stimuli.There are two solutions to this issue. Either you do additional experiments and show that a reasonable set of features does not explain differences between the two tasks, or you revise the manuscript to remove all language about social vs non-social and acknowledge that the differences between the tasks could be entirely driven by differences in stimulus features.

This concern is addressed in Public Response 1. However, in brief, we hope we have allayed the Reviewer’s concern by (1) using a prior data set with a similar task in which the social and nonsocial stimuli’s complexity are visually matched to demonstrate there are still learning differences, and (2) revising the manuscript (page 15-16, lines 376-380) to acknowledge that in the present work there are differences in stimulus complexity between the two conditions, which may contribute, in part, to these observed learning differences. Further, while we acknowledge that stimulus complexity could produce better decoding of faces during choice (Figure 5B), our RSA findings illustrate that the fidelity of state representations are yoked to individual differences in credit assignment across tasks (social and nonsocial contexts). This suggests that while stimulus complexity may play *some* role in the fidelity of these state representations, it seems that how well an individual is able to assign credit is more closely related to the nature of these representations.

5. Your modeling approach is designed to test (among other things) whether working memory or credit assignment mechanisms better account for the behavioral data. Given that these mechanisms are not mutually exclusive, it would be important to include a model that captures both mechanisms.

This point is addressed in point 2.

6. I think using the term 'state' when referring to 'stimulus' is misleading. States are generally thought to be abstract and not tied to specific stimulus configurations. To dissociate states and stimuli, you'd need an experiment where the same state is evoked by different stimuli. Your design actually allows you to do that if you assume that faces and corresponding bandits evoke the same state.

This point is addressed in point 3.

7. I may be wrong here, but if I understand your description of the cross-timepoint RSA in the method section correctly, I wonder whether this analysis could still be confounded by the temporal proximity of choice and feedback phase. As far as I understand, you computed the similarity of choice and feedback phases within odd trials and separately within even trials. Then you computed the matrix product between the odd and even trial neural RDMs. Couldn't any signals (including those driven by vascular effects or noise) that linger from the choice phase into the feedback phase explain the cross-phase similarity within the odd and even trials? It seems it would be better to compute two neural RDMs such that the choice phase comes from odd trials and the feedback phase comes from even trials (and vice versa), and then average the two RDMs. This way, the similarity between the choice and feedback phase is less likely to result from autocorrelations

Thank you for the thorough reading of our methods and this technically thoughtful point—your suggestion is exactly what we did! We apologize that our description of the cross-timepoints RSA was not clear which led to a misunderstanding of the analysis. To clarify the procedure we implemented in Figure 6A-C: we extracted β coefficients from each voxel on all odd trials during choice with each partner and bandit (separately) and during feedback with each partner and bandit (separately). We did the same thing for even trials.

We then correlated the average activation patterns of the odd trials for choice with the even trials for feedback (for each stimulus at each timepoint). We did the same for even trials for choice and odd trials for feedback. This allows us to compare mean activation patterns on even trials during choice (choiceEven) compared to mean activation patterns on odd trials during feedback (feedbackOdd), and vice versa for feedbackEven trials and choiceOdd trials. Because we never crossed choice and feedback with activations from the same trial, it is unlikely that our effects are biased by temporal autocorrelation. Thank you for giving us the opportunity to clarify this, and we have revised the Methods section on page 23, lines 640-666 to reflect the analyses steps in more detail.

8. How were the ROIs selected? I assume there were more areas that represented stimulus identity, no?

We used a data-driven approach to identify ROIs. We created a task-combined *t*-map of all the regions representing stimulus identity (corrected for multiple comparisons with a permutation testing procedure), and then selected the top 10 peak activations from these significant clusters. We then limited the size of our ROIs to be no greater than a searchlight (9mm radius). Our procedure for ROI selection is detailed in the Methods section (page 22, lines 606-623).

Reviewer #2 (Recommendations for the authors):In this manuscript the authors use a combination of reinforcement learning modelling and fMRI to study the fidelity of state representations in the frontal cortex and whether these representations predict the ability of (individual) participants for credit assignment.Strengths:1. This study provides a nice combination of reinforcement learning modelling and fMRI studies, which enabled the authors to link computational principles with neuronal representations.3. The experimental paradigm is also interesting, contrasting social and non-social tasks. It suggests very interesting differences between the two in terms of investment, but also in terms of neural representations.4. Finally, this study might make substantial advances in our understanding of individual differences in terms of credit assignment, a critical part of learning and adaptability. However, this is not entirely clear (see below).Weaknesses:1. The manuscript could present some of the results in a more gradual way, that makes it accessible to a general reader. I also find that there are generally long relatively complex sentences that make it hard to follow. For example: "We leverage these potential differences to evaluate whether the brain adaptively adjusts the fidelity of state representations across contexts to mediate the selectivity of causal learning.".

Thank you for the feedback on the manuscript’s clarity. We carefully revised the manuscript in many places where there were overly complicated and technical sentences. We hope that the revised manuscript is more accessible, and Oriel wants you to know that after 6 years she feels seen.

2. There is in my view a need to clarify what is meant by credit assignment (CA). For example, the way the models are described in the main text make it seem that some models perform credit assignment whereas others do not. From what I can see all these models have to perform some form of credit assignment, as they all atribute credit to model parameters/states, which I think would also make model 1 a CA model.

This is valuable insight and we agree with the conceptual point—that in principle, all of our models capture the assignment of credit in some way (i.e., in the base Rescorla-Wagner model, CA is assumed to be perfect, corresponding to a CA parameter value of 1). However, we also want to avoid confusing readers about what we aimed to test though model comparison. To this end, we have edited our modeling section of the main manuscript (page 7, lines 173-189), and our Methods section (page 19-20, lines 497-527) to make it clear that we tested a set of RL models, some of which include an explicit CA parameter aimed to evaluate CA precision, and some of which do not, but which still implicitly assume that credit is being assigned. We hope that these revisions address any potential confusion.

3. One of the key points made by the authors is that individual differences are driven by the strength of neural representations, and not by the magnitude of learning signals. Unfortunately, I fail to see how this conclusion is supported by the analysis of the data. I believe that this is interpretation builds on their correlation analysis between the representations and the model fits (Figure 5D and 6C). However, the model itself contains an explicit learning signal (δ: prediction error), so it is not clear to me how can the authors disentangle the neuronal representations from the learning signal using this analysis.

Thank you for bringing this to our attention and we apologize that we were not more thorough in our manuscript. It is worthy of note that, as the reviewer says, learning in our model still requires learning signals. Our claim is that individual differences in learning stem not from differences in the magnitude of these signals, but in what they are attributed to. We did not initially include the prediction error/learning rate analysis because we were concerned about the manuscript’s breadth and wanted to avoid taxing readers. However, we agree that to make this claim in our significance statement we need to include the relevant analyses, so have added an additional section to the main text on page 14. In these analyses we use parametric modulation to show that prediction error signaling does not differ between the two tasks, and the learning rate from the model does not predict the format of neural representations. We hope that these analyses are sufficient to support our claim that the magnitude of learning signals do not seem to account for learning differences in our task.

4. The model and some of the methods are not described in enough detail. For example, it is not stated what some of the parameters are. Although the models used appear to be standard in the field, no links/citations to classical RL models are made.

We apologize that we did not provide sufficient detail or citations about which RL model we used. We used the Rescorla-Wagner (RW) learning rule in the baseline model (Rescorla, 1972). The decay model was previously described in (Collins & Frank, 2012), in which a decay parameter gradually adjusts learned values back to initial ones (i.e., the prior), proportionally to the degree of forgetting. We developed the credit assignment model which incorporates a CA parameter that adjusted the RW learning rule such that prediction errors can also influence the expected value of irrelevant states. We have revised the computational model sections of the manuscript on page 7 and our methods section (page 19-20, lines 497-527), to provide more explicit details and citations.

Claims:Several claims are supported by the data/analysis, but it is not clear to me if one of the central claims is (see one of the weakness points above).1. One of the potentially very interesting points made by the authors is about what causes individual differences in terms of CA. However, I fail to see how they can use the CA fit without implicitly also considering learning signals -- learning signals are an implicit part of their model (i.e. there is a δ = prediction error). Not sure how this can be addressed, but maybe there is something I'm missing and this is not a problem at all? On a related point, I failed to see a discussion on this very important part of the manuscript, discussing what would be the key contribution of the paper.

We appreciate the feedback (and the enthusiasm about our findings!) and have included additional analyses on page 14 as outlined in public response 3. As you correctly note, in the current manuscript we do not attempt to identify or explain causal sources for individual differences in credit assignment. In this paper we limited the scope of what we were testing and simply focused on the finding that people vary in credit assignment precision, this variability is reflected in the format of PFC state representations, and we provide a mechanistic explanation for why this may be beneficial for learning. However, as a future direction, we are interested in exploring the possibility that PEs may help explain how different credit assignment policies are learned over time and contribute to individual differences in credit assignment precision. We think that these questions warrant their own thorough investigation in separate manuscript with new data.

2. The text could be improved by making more smoother/gradual descriptions of the results/ideas. For example, I find that there is a lot of information in the final part of the section "Although these outcomes should not inform choices on the current trial given the generative task structure…" (until the end). To make it more accessible to the general reader I suggest that you present this information more gradually and provide intermediate (brief) summaries of what it means.

This concern is addressed in point 1.

3. So I think it would be better to clarify that these are all different variants of CA models. This does not change your interpretation in any way but would make the story more clear. Also, to follow the modelling more easily I would suggest giving specific names to each model like you already do for the v-CA model. I note that this is done in Figure S2, but should also be included in the main text, and in the methods when referring to the methods, for clarity. This would also make the link with Figure 3 easier to follow. For example, at the moment it's unclear exactly to which model Figure 3b and c refer.

This is a great suggestion to label each model, and we have revised the modeling section on page 7 to make the models names more explicit. We have also revised the caption for Figure 3B-C to improve clarity that the plotted CA fits are from the V-LR, V-CA model, and have included model names throughout the manuscript where applicable.

4. On page 7 and Figure 3 you use the term "spread" again, but this was only referred to very early on, so at this point appears out of place. Would be important to highlight what this means and how the models capture this spread. Also, in Figure 3 the spread is contrasted with CA precision, given that these two elements are critical for understanding Figure 3, it would be important to clearly define them in the main text before introducing the results of Figure 3. Also, in Figure 3 are the credit matrices obtained by running the models? If so, how? Clarify.

Thank you for the suggestion. We have revised both the introduction (page 2, lines 41-45) and results (page 7, lines 175-178) to make the point clear that spread and CA precision are on opposite ends of the spectrum. In the modeling section on line 177, we reintroduce the term “spread” and define it again to readers. As to your second point, the credit matrices shown in Figure 3 are only hypothetical matrices to illustrate the logic that increased precision should in theory promote increased differentiation between stimuli, and increased spreading would result in increased similarity between stimuli and, thus more confusability. We have revised the figure caption (page 8, line 195-209) to clarify that these matrices are for illustration purposes only.

5. One possible prediction for the neuronal representations of PE found in PFC is that they should weaken over learning if they underlie the learning signal. As this form a critical component of any credit assignment/learning model, it would be interesting/important to explore this. I imagine this could be easily tested with your data by analysing how the PE-specific representations change over learning.

This is a great suggestion, and we assume the reviewer means the neural state representations that we show in Figure 5. The point is well taken that state representations could weaken or strengthen if they underlie an instructive learning signal (e.g., task demands). As one avenue to explore in future work, we are very intrigued by the possibility that state representations are dynamically reorganized over the course of learning and then exploited by the control network to execute optimal choice. However, we do we want to stretch the scope of what we are hoping to achieve in the current paper.

6. There is some lack of clarity in the description of the model/analysis: (i) The baseline RL model, seems to be exactly the Rescorla-Wagner rule. If so please refer to it as so. (ii) The AIC is usually given by: AIC = – 2ln(L) + 2k, not AIC = – 2ln(L) – 2k, as given in the methods. Please clarify. (iii) What is the prior in the decay model? It does not appear to be given. (iv) The Q variable is usually reserved for Q-learning in RL, which models explicitly the value of state-action pairs, but this does not appear to be the case here. Instead, to be consistent with the literature, I suggest that you use the variable V. It's important that the authors ensure that all the details are given, and parameters described.

The baseline RL model is the Resclora-Wagner learning rule as the reviewer correctly notes—and we have edited the manuscript to make this explicit (page 7, line 181; page 19, line 497). In terms of AIC, we fit our models by minimizing the negative sum of the Bayesian loglikelihood, as opposed to standard loglikelihood values, so we simply signed flipped the AIC equation which is why the original equation reported in the manuscript subtracted the penalty term. However, we agree with the reviewer that it is better to be consistent with established equations used in the field and have implemented this correction on page 20, line 535, in the Methods section. The prior in the Decay model was a free parameter that estimated the initial value in the Q-matrix (now renamed V-matrix). We have clarified this on line page 19, lines 498-501 and on page 20, line 521-524 in Methods. Thanks for the suggestion regarding the Q-value, we have changed it to V throughout.